# Continual Learning for Long-Tailed Recognition

## Abstract

We propose Continual Learning for Long-Tailed Recognition (CLTR), a framework that employs standard off-the-shelf Continual Learning (CL) methods for addressing Long-Tailed Recognition (LTR) problems, by first learning the majority classes (Head) followed by learning of the minority classes (Tail), without forgetting the majority. To ensure that our method is theoretically sound, we first prove that training a model on long-tailed data leads to weights similar to training the same learner on the Head classes. This naturally necessitates another step where the model learns the Tail after the Head in a sequential manner. We then prove that employing CL can effectively mitigate catastrophic forgetting in this setup and thus improve the model's performance in addressing LTR. We evaluate the efficacy of our approach using several standard CL methods on multiple datasets (CIFAR100-LT, CIFAR10-LT, ImageNet-LT, and Caltech256), showing that CLTR achieves state-of-the-art performance on all the benchmarks. Further, we demonstrate the effectiveness of CLTR in the more challenging task of class-incremental LTR, surpassing the state-of-the-art methods in this area by notable margins. Lastly, extensive sensitivity analyses and detailed discussions are provided to further explore the underlying mechanisms of CLTR. Our work not only bridges LTR and CL in a systematic way, but also paves the way for leveraging future advances in CL methods to more effectively tackle LTR problems.

## 1 Introduction

Data in real-world scenarios often exhibits long-tailed distributions (Buda et al., 2018; Reed, 2001; Zhang et al., 2023; Fu et al., 2022), where the number of samples in some classes (Head set) is significantly larger than the number of samples in other classes (Tail set). This imbalance can lead to sub-optimal performance in deep learning models. This problem is known as Long-Tailed Recognition (LTR), which can be described as training a model on highly imbalanced data and attempting to achieve high accuracy on a balanced test set (Zhang et al., 2023).

Given that the size of the Head set is substantially larger than the Tail set, samples from the Head generally dominate the loss and determine the gradient. Consequently, samples from the Tail are less impactful, leading to strong performance in Head classes but a significant decline in the performance of the Tail classes (Alshammari et al., 2022). Numerous studies have sought to mitigate this issue by balancing training data through over-sampling the Tail classes (sample-wise balancing) (Chawla et al., 2002; Estabrooks et al., 2004; Feng et al., 2021). Alternatively, feature extractors have been trained using the Head set and adapted through transfer learning to be used for the Tail classes (Liu et al., 2019; Wang et al., 2017; Zhong et al., 2019; Jamal et al., 2020). Another approach has been to regularize the loss or gradient selectively, depending on the size of the class set (loss-wise balancing) (Cao et al., 2019; Cui et al., 2019; Tang et al., 2020). Weight balancing has been proposed as a method for penalizing excessive weight growth during training, thus forcing per-class weight norms to maintain more uniform magnitudes (Alshammari et al., 2022). However, both sample-wise and loss-wise balancing methods lead to increased sensitivity to variations in the tail (Wang et al., 2021c). It has also been shown that these methods may compromise the representational capability of the deep features learned by the model (Zhou et al., 2020). To address these issues, multi-stage training has recently been proposed as a viable approach to this problem (Zhou et al., 2020; Zhang et al., 2022). However, these solutions often rely on an ensemble of multiple experts or backbones to allow effective training of both

Head and Tail sets, as the use of a single model for multi-stage training would likely result in catastrophic forgetting.

To address this, we propose a simple yet novel framework called **C**ontinual **L**earning for Long-**T**ailed **R**ecognition (CLTR), which formulates LTR as a sequential learning problem where the Head classes are learned first, followed by the Tail classes. In this framework, our method draws on the benefits of CL to alleviate catastrophic forgetting and retain both Head and Tail information effectively. To ensure that our approach is theoretically well-grounded, we first prove that training a model on an LTR dataset leads to similar weights as training the same model solely on the Head. This naturally leads to the need for an additional step where the Tail classes are further learned by the model, i.e., a sequential learning of the Head followed by the Tail. Next, we prove that CL can effectively mitigate catastrophic forgetting in this setup and allow for effective learning of the Tail without forgetting the Head. We validate our theory and the efficacy of CLTR using five datasets, MNIST-LT, CIFAR100-LT, CIFAR10-LT, ImageNet-LT, and Caltech256. First, we use the toy MNIST-LT dataset and show that the actual distance between weight vectors when trained on either the Head or the entire dataset aligns closely with our theoretical predictions. Next, to further assess the efficacy of CLTR, we employ a range of CL methods in our framework and evaluate the performance on LTR benchmarks, namely CIFAR100-LT, CIFAR10-LT, and ImageNet-LT, with varying imbalance factors. The results indicate that CLTR consistently achieves either the best or second-best performance across all benchmarks, affirming its viability as a long-tailed classifier. We then compare the performance of our model to recent works on Long-Tail class-incremental Learning (LT-CIL), and show that CLTR outperforms the state-of-the-art methods. Finally, we offer a discussion on the implications of the proposed perspective for LTR and the limitations of our study.

Our contributions are as follows: (**1**) We propose and prove a theorem that sets an upper bound on the distance between weights obtained when training a learner on different partitions of an imbalanced dataset, under the assumption of strong convexity of the loss function. This bound is inversely proportional to the imbalance factor and proportional to the strong convexity of the loss function. (**2**) Building on this theorem, we introduce a new approach that employs CL solutions for the LTR problem using a sequential learning framework. To support this approach, we prove the effectiveness of CLTR in reducing the loss when learning the Head and Tail sets sequentially. (**3**) We substantiate our method through comprehensive experiments that show the effectiveness of our CLTR framework in addressing LTR and LT-CIL problems. Our results indicate that using CLTR leads to state-of-the-art performances in both problem setups.

## 2 Related Work

**Long-Tailed Recognition.** Real-world datasets often exhibit imbalanced distributions, with some classes appearing more frequently than others. Training a model on such imbalanced data can result in poor performance on the rare classes. LTR addresses this issue by enabling models to perform well on both Head and Tail classes (Cao et al., 2019). LTR approaches can be broadly categorized into three primary groups: *data distribution re-balancing*, *class-balanced losses*, and *transfer learning from Head to Tail* (Kang et al., 2019). Data distribution re-balancing techniques include over-sampling the Tail (Chawla et al., 2002; Han et al., 2005), under-sampling the Head (Drummond et al., 2003), and class-balanced sampling (Shen et al., 2016; Mahajan et al., 2018). Class-balanced loss approaches modify the loss function to treat each sample differently, e.g., including class distribution-based loss (Cao et al., 2019; Cui et al., 2019; Huang et al., 2019), focal loss (Lin et al., 2017), and Bayesian uncertainty (Khan et al., 2019). Additionally, transfer learning techniques leverage features learned from the Head to improve learning on the Tail (Yin et al., 2019; Liu et al., 2019). More recently, the limitations of class re-balancing have been discussed and the Bilateral-Branch Network (BBN) was proposed to improve representation learning (Zhou et al., 2020). This method addresses the training of the encoder and classifier separately through a novel cumulative learning strategy that initially focuses on universal patterns before progressively concentrating on the Tail. The RoutIng Diverse Experts (RIDE) model is introduced to enhance LTR by reducing model variance (Wang et al., 2021c). Finally, the assumption that the test set distribution is always uniform is challenged and test-agnostic long-tailed recognition is introduced (Zhang et al., 2022). The authors discuss that self-supervised learning facilitates universal feature learning, improving performance on test sets with unknown distribution. To this end, they introduce a new method that trains multiple experts on a long-tailed dataset to manage various class

distributions and uses self-supervision at test time to combine these experts for unknown class distributions. Although numerous prior works have addressed LTR, few provide a mathematical analysis of the training process using imbalanced data (Ye et al., 2021; Francazi et al., 2023). These works demonstrate that the Head is learned more quickly than the Tail, primarily focusing on the training dynamics. In contrast, our theoretical analysis studies the convergence point of training within the LTR framework.

As discussed, some LTR solutions fall into the category of multi-stage training (Zhou et al., 2020; Zhang et al., 2022). Our work here extends this by first presenting a formal framework in which LTR is formulated as a sequential problem. Along with the theoretical foundations that describe why sequential learning is particularly well-suited for LTR, we also identify the key factors that influence the success of these methods. Subsequently, we propose that CL be used as a viable and highly effective solution for LTR. This allows us to draw from a rich pool of prior work on CL, which unlike existing multi-stage learning solutions to LTR, use only a single network throughout the training.

**Continual Learning.** CL addresses the challenge of adapting a deep learning model to new tasks (e.g., new classes or distributions) while maintaining performance on the previously learned tasks. The main challenge to address by CL methods is the mitigation of catastrophic forgetting, i.e., forgetting the previous tasks as the new tasks are learned. CL methods are typically grouped into three categories: *expansion-based*, *regularization-based*, and *memory-based approaches*. *Expansion-based* CL methods utilize a distinct subset of parameters for learning each task (Sarwar et al., 2019; Li et al., 2019; Yoon et al., 2020). *Regularization-based* techniques penalize significant changes in crucial network parameters (relative to previous tasks) by incorporating a regularization term in the loss function (Saha et al., 2020; 2021; Farajtabar et al., 2020; Kirkpatrick et al., 2017; Li & Hoiem, 2017). *Memory-based* approaches employ a replay memory to store a limited number of samples from previous tasks, which are then used in future training to minimize forgetting (Riemer et al., 2018; Chaudhry et al., 2019; Shim et al., 2021). FOSTER uses a two-stage paradigm to dynamically expand and compress modules when learning new tasks (Wang et al., 2022). Task-id Prediction based on Likelihood Ratio (TPL) is proposed in (Lin et al., 2024a) for class-incremental Learning. This method utilizes likelihood ratios for task-id prediction by leveraging available replay data and task-specific models trained within a shared network. It thus overcomes the challenge of task identification in the absence of explicit task identifiers at test time. More recently, gradient surgery has been employed for addressing CL where the gradient from the new task is projected to the orthogonal direction of the previously learned tasks to ensure learning the new task does not impact the previous task (Saha et al., 2020; Saha & Roy, 2023). These methods achieve state-of-the-art performance on CL benchmarks.

**Long-Tailed Class-Incremental Learning.** Few prior works have attempted to address the problem of class-incremental learning when the data is heavenly imbalanced. A novel replay method called Partitioning Reservoir Sampling (PRS) is proposed in (Kim et al., 2020). This method dedicates a sufficient amount of memory to tail classes in order to avoid catastrophic forgetting in minority classes. In (Liu et al., 2022a), this problem is addressed in two different setups, ordered and shuffled. In the ordered scenario the number of samples in each new task is less than in previous tasks, while in the shuffled scenario, the size of classes is completely random. They propose a two-stage learning method utilizing a learnable weight scaling layer for reducing the bias due to data imbalance. Finally, in (Liu et al., 2022b), OLTR++ is proposed which is a unified algorithm that integrates imbalanced classification, few-shot learning, open-set recognition, and active learning through dynamic meta-embedding and memory association. Note that none of the above works attempt to employ CL as a solution for LTR scenarios.

## 3 Proposed Approach

### 3.1 Training on Long-Tailed Distributions

In this section, we first define the LTR problem and then analyze the behavior of a model when trained on long-tailed distributions to provide a theoretical basis for our proposed CLTR framework. Let's consider the input space to be $\mathbb{R}^d$, where each input is represented by $x_i$, and the label space is $\{1, \ldots, k\}$, where each label is denoted by $y_i$. Let $\mathcal{D}$ denote the training set containing samples $(x_i, y_i)$. $\mathcal{D}_c$ is a subset of $\mathcal{D}$ where $\mathcal{D}_c = \{(x_i, y_i) \in \mathcal{D} \mid y_i = c\}$ and $|\mathcal{D}_c|$ represents its cardinality. Without loss of generality, let the

classes be ordered by their cardinalities such that $|\mathcal{D}_i| \geq |\mathcal{D}_j|$ for all $i < j$. Following (Hong et al., 2024), let $\mathcal{D}_H$ and $\mathcal{D}_T$ represent the subsets of $\mathcal{D}$ corresponding to the Head set and Tail set, respectively as $\mathcal{D}_H = \{(x_i, y_i) \in \mathcal{D} : y_i \leq c_k\}$ and $\mathcal{D}_T = \{(x_i, y_i) \in \mathcal{D} : y_i > c_k\}$, where $c_k$ denotes how many classes belong to each set. As a result, every class in the Head has more samples than any class in the Tail. The loss function over $\mathcal{D}_c$ is defined as $\mathcal{L}(\mathcal{D}_c, \theta) = \frac{1}{|\mathcal{D}_c|} \sum_{i=1}^{|\mathcal{D}_c|} \ell((x_i, y_i), \theta)$, where $(x_i, y_i) \in \mathcal{D}_c$. Note that $\ell((x_i, y_i), \theta)$ is the loss of each individual sample. For brevity, we will henceforth use notations $\mathcal{L}_{\mathcal{D}_c} = \mathcal{L}_{\mathcal{D}_c}(\theta) = \mathcal{L}(\mathcal{D}_c, \theta)$ and $\ell((x_i, y_i)) = \ell_{(x_i, y_i)}(\theta) = \ell((x_i, y_i), \theta)$.

LTR aims to address the challenge of learning from highly imbalanced data. This occurs when the training data $\mathcal{D}$ contains more samples in some classes (the Head set $\mathcal{D}_H$) and fewer in others (the Tail set $\mathcal{D}_T$). The imbalance factor IF quantifies the severity of this issue in a dataset, defined as:

$$\text{IF} = \frac{|\mathcal{D}_{c^{\max}}|}{|\mathcal{D}_{c^{\min}}|}, \tag{1}$$

where $c$ represents the class index, $|\mathcal{D}_c|$ denotes the cardinality of each class, $c^{\max} = \arg\max |\mathcal{D}_c|$, and $c^{\min} = \arg\min |\mathcal{D}_c|$, such that $\mathcal{D}_{c^{\max}} \in \mathcal{D}_H$ and $\mathcal{D}_{c^{\min}} \in \mathcal{D}_T$. Now we formally define a long-tailed dataset and the LTR problem in the following:

**Definition 3.1.** *A dataset is deemed* long-tailed *when* $|\mathcal{D}_{c^{\max}}| \gg |\mathcal{D}_{c^{\min}}|$ *or, in other words,* IF $\gg 1$. *When a model is trained on such a dataset and its performance is assessed on a test set where each class $c$ has the same number of samples (i.e. $|\mathcal{D}_c| = \kappa$ for each class $c$ within the test set where $\kappa$ is a constant number), the problem is referred to as* Long-Tailed Recognition.

**Assumption 3.2.** *We initially assume that all head classes are of size $|\mathcal{D}_H|$, and all tail classes are of size $|\mathcal{D}_T|$, with $|\mathcal{D}_H| \gg |\mathcal{D}_T|$. These assumptions will be relaxed later in Assumption 3.7. The model is a logistic regression classifier with parameters $\theta$ trained with regularized crossed-entropy loss which is a combination of cross-entropy loss and an additional $L^2$ regularization term $\frac{\mu}{2}\|\theta\|^2$ that prevents weights from growing excessively.*

Assumption 3.2 helps simplify the derivation of the following theoretical analysis. However, our framework shows strong performances even when all the constraints in this Assumption are relaxed, as presented in Section 4.2. We now introduce Theorem 3.3 demonstrating the relationship between the weights of the model when it is trained solely on the head as well as on the entire dataset.

**Theorem 3.3.** *Given Assumption 3.2, if a model with parameter vector $\theta$ is trained in an LTR setting (Definition 3.1), then,*

$$\|\theta^* - \theta_H^*\|^2 \leq \frac{4\delta}{\mu_H + \mu}, \tag{2}$$

*where $\theta^*$ represents the parameter vector obtained after training, $\theta_H^*$ denotes the parameter vector when the model is trained solely on the Head set, $\delta$ is the maximum difference between the loss of the learner using the entire dataset and the Head set $(|\mathcal{L}(\mathcal{D}) - \mathcal{L}(\mathcal{D}_H)| \leq \delta)$ for any value of $\theta$, and $\mu_H$ and $\mu$ are the strong convexity parameters of the loss calculated on the Head set and the entire dataset, respectively.*

To prove this theorem, we first introduce Lemma 3.4, which shows that when the difference between two strongly convex functions is bounded, their minimizers also reside in a bounded neighborhood of each other.

**Lemma 3.4.** *If $|f(x) - g(x)| \leq \delta$ and both $f(x)$ and $g(x)$ are strongly convex, then:*

$$\|x_g - x_f\|^2 \leq \frac{4\delta}{\mu_f + \mu_g}, \tag{3}$$

*where $x_g$ and $x_f$ are $\arg\min f(x)$ and $\arg\min g(x)$, respectively.*

For the full proof of Lemma 3.4, see Appendix A.1.

*Proof of Theorem 3.3.* The model is trained on the entire dataset $\mathcal{D}$ by minimizing the loss function $\mathcal{L}$ defined as:

$$\mathcal{L}(\mathcal{D}) = \frac{1}{|\mathcal{D}|} \left( \sum_{(x_i, y_i) \in \mathcal{D}_H} \ell((x_i, y_i)) + \sum_{(x_i, y_i) \in \mathcal{D}_T} \ell((x_i, y_i)) \right), \tag{4}$$

Using $\mathcal{L}(\mathcal{D}_H) = \frac{1}{|\mathcal{D}_H|} \sum_{(x_i,y_i)\in\mathcal{D}_H} \ell((x_i,y_i))$ and $\mathcal{L}(\mathcal{D}_T) = \frac{1}{|\mathcal{D}_T|} \sum_{(x_i,y_i)\in\mathcal{D}_T} \ell((x_i,y_i))$, we can derive:

$$\mathcal{L}(\mathcal{D}) = \frac{|\mathcal{D}_H|}{|\mathcal{D}|} \mathcal{L}(\mathcal{D}_H) + \frac{|\mathcal{D}_T|}{|\mathcal{D}|} \mathcal{L}(\mathcal{D}_T). \tag{5}$$

Now we define $\gamma = \frac{|\mathcal{D}_H|}{|\mathcal{D}|}$. Since $|\mathcal{D}| = |\mathcal{D}_H| + |\mathcal{D}_T|$, we can derive that $1 - \gamma = \frac{|\mathcal{D}_T|}{|\mathcal{D}|}$. Plugging $\gamma$ into Eq. 5 yields:

$$\mathcal{L}(\mathcal{D}) = \gamma \mathcal{L}(\mathcal{D}_H) + (1 - \gamma)\mathcal{L}(\mathcal{D}_T). \tag{6}$$

Given that IF $= \frac{|\mathcal{D}_H|}{|\mathcal{D}_T|}$, hence $\gamma = \frac{\text{IF}}{1+\text{IF}}$, which falls within the range of $[0.5, 1)$. Since based on Definition 3.1, IF $\gg 0$ in LTR, we can conclude that the value of $\gamma$ approaches one. Consequently, $\mathcal{L}(\mathcal{D})$ approaches $\mathcal{L}(\mathcal{D}_H)$ for all $\theta$ values. Let $\delta$ be defined as the maximum difference of the losses:

$$|\mathcal{L}(\mathcal{D}) - \mathcal{L}(\mathcal{D}_H)| < \delta. \tag{7}$$

From Eq. 6, it follows that $\lim_{\text{IF}\to\infty} \delta = 0$.

Following Assumption 3.2, the loss function can be formulated as:

$$\mathcal{L}(\mathcal{D}, \theta) = -\frac{1}{N} \sum_{i=1}^{N} y_i \log\left(P(f(\theta, x_i))\right) + \frac{\mu}{2}\|\theta\|^2, (x_i, y_i) \in \mathcal{D}. \tag{8}$$

where $P(.)$ is the softmax function, $f(\theta, x_i)$ is the output of the logistic regression with inputs $x_i$ and parameters $\theta$, and $\mu$ denotes the coefficient of the regularization term. The value of this training hyper-parameter is determined by the user, usually through hyper-parameter tuning, grid search, or similar approaches. This loss function is employed because it is highly effective for the LTR problem, improving generalizability by reducing overfitting and achieving state-of-the-art performance when dealing with LTR scenarios (Alshammari et al., 2022). As our model is assumed to be a logistic regression classifier, the Hessian of the cross-entropy loss, $\nabla_\theta^2 \mathcal{L}_{CE}(\mathcal{D}, \theta)$, is positive semi-definite, where $\nabla_\theta^2$ denotes the Hessian with respect to $\theta$. Adding a regularization term with a coefficient $\frac{\mu}{2}$ results in a positive definite Hessian matrix with a lower bound of $\mu$. Therefore, the Hessian matrix satisfies $\nabla_\theta^2 \mathcal{L}(\mathcal{D}, \theta) \succeq \mu I$. Consequently, the eigenvalues of the Hessian matrix are bounded below by $\mu$, ensuring that $\mathcal{L}$ is a strongly convex loss function, where $\mu$ represents the extent of the convexity. From the definition of strong convexity (Sherman et al., 2021), it therefore follows that:

$$\mathcal{L}(x_1) \geq \mathcal{L}(x_2) + \nabla\mathcal{L}(x_2)^T(x_1 - x_2) + \frac{\mu_{\mathcal{L}}}{2}\|x_1 - x_2\|^2, \tag{9}$$

where $\mu_{\mathcal{L}}$ is the strong convexity parameter. A more detailed discussion of the strong convexity of the loss function, its properties, and the relevance of the proposed theoretical analysis to the LTR problem is provided in Appendix B.

Applying Lemma 3.4 to Eqs. 7 and 9 yields:

$$\|\theta^* - \theta_H^*\|^2 \leq \frac{4\delta}{\mu_H + \mu}, \tag{10}$$

where $\theta^*$ and $\theta_H^*$ are $\arg\min\mathcal{L}$ and $\arg\min\mathcal{L}_H$, respectively. $\qquad\square$

As a result, when the model is trained on a long-tailed dataset, the network parameter $\theta$ converges to a point close to the weights of the model when it was only trained on the Head set $\theta_H$. It is worth mentioning that if the same coefficient for the regularization term $\mu$ is used for both $\mathcal{L}$ and $\mathcal{L}_H$, the lower bound in Eq. 2 can be further simplified to $\frac{2\delta}{\mu}$. To further analyze the training of the model under the LTR scenario, let us relax the assumption on the loss function and assume that the model is just using cross-entropy loss without the regularization term. This leads to the following remark:

**Remark 3.5.** *The upper bound of the distance between a learner's parameters when trained on the entire dataset, and the parameters of the same learner solely trained on the Head set, can be calculated as:*

$$\|\theta^* - \theta_H^*\|^2 \leq \frac{4\delta}{\lambda + \lambda_H}, \tag{11}$$

*where $\lambda$ and $\lambda_H$ are the minimum eigenvalues of the hessian matrices of $\mathcal{L}(\mathcal{D})$ and $\mathcal{L}(\mathcal{D}_H)$, respectively*

*Proof of Remark 3.5.* We first show through Lemma 3.6 that if the difference between two convex functions is bounded, then the distance between their minimizers can also be bounded.

**Lemma 3.6.** *If $|f(x) - g(x)| \leq \delta$ and both $f(x)$ and $g(x)$ are strictly convex, then:*

$$\|x_f - x_g\|^2 \leq \frac{4\delta}{\lambda_f + \lambda_g}, \tag{12}$$

*where $x_g$ and $x_f$ are $\arg\min f(x)$ and $\arg\min g(x)$, and $\lambda_f$ and $\lambda_g$ are the minimum eigenvalues of the hessian matrices of $f(x)$ and $g(x)$, respectively.*

The full proof of Lemma 3.6 is provided in Appendix A.2. Since the loss is now unregularized cross-entropy, the loss function is strictly (but not strongly) convex (i.e. $\nabla^2 \mathcal{L}(\mathcal{D}, \theta) \geq 0$.) Hence, by applying Lemma 3.6 and Eq. 7 we conclude Eq. 11, which completes the proof. □

To ensure that the upper bound expressed by Remark 3.5 is limited and approaches zero when $\delta \to 0$, the minimum eigenvalues of the Hessian of both loss functions should have lower bounds, which is again another definition of strong convexity and verify our finding in theorem 3.3.

Theorem 3.3 assumes that there is only one Head and one Tail in the dataset, which is not the case in many real-world datasets. So we are relaxing Assumption 3.2 as follows:

**Assumption 3.7.** *Building upon Assumption 3.2, we modify the distribution of classes. Specifically, we no longer assume that all classes are of equal size within the Head and Tail sets, respectively. Instead, the model accommodates a scenario where the number of samples in the Head classes can differ from each other and the same applies to the Tail, without specifying the relationship in size between $|\mathcal{D}_H|$ and $|\mathcal{D}_T|$.*

Under the relaxed assumption where the size of the classes within Head and Tail sets differ, these sets can be each further partitioned into their own distinct Head and Tail subsets. While each individual partition remains imbalanced, we continue to subdivide them until: (1) $|\mathcal{D}^i| >> |\mathcal{D}^j|$ for $i < j$, and (2) $\mathrm{IF}_{\mathcal{D}^i} \not\gg 1$ for all partitions $\mathcal{D}^i$. In this scenario, there is no long-tailed partition of the data. Theorem 3.8 extends Theorem 3.3 to address this scenario for any number of partitions.

**Theorem 3.8.** *Following Assumption 3.7, we divide the dataset $\mathcal{D}$ into $n$ partitions. Let a subset of $m \leq n$ partitions be $\bigcup_{i=1}^{m} \mathcal{D}^i \subseteq \mathcal{D}$, with the largest partition being $\mathcal{D}^a$, i.e. $a = \arg\max_i |\mathcal{D}^i|, i \in [1, m]$. Then, the weights $\theta^*_{\bigcup \mathcal{D}^i}$ obtained from training the model on $\bigcup_{i=1}^{m} \mathcal{D}^i$ will always be in a bounded neighborhood of the weights $\theta^*_{\mathcal{D}_a}$ obtained from training on the largest subset $\mathcal{D}^a$.*

*Proof Sketch.* We start by dividing the dataset into multiple partitions each substantially larger than the previous one. We then apply Theorem 3.3 on the two largest subsets and find the upper bound for the weight differences. We then consider the aggregation of these two subsets as the new 'largest' subset and apply Theorem 3.3 to this 'largest' subset and the next largest partition to find a new upper bound. Repetitively applying Theorem 3.3 allows us to calculate an ultimate upper bound for the weight difference when training on the largest subset versus the entire dataset. The formal proof is provided in Appendix A.3.

## 3.2 Continual Learning for Long-Tailed Recognition

Let us assume an LTR problem and a learner with a set of parameters denoted as $\theta$ (recall definition 3.1). Initially, the learner is trained on a highly imbalanced dataset $\mathcal{D}$, as shown in Fig. 1, where $\theta_i$ is the initialized model in the weight space. Owing to the larger number of Head samples in each iteration, they dominate the evolution of the gradients (Eq. 5), resulting in a learner that performs significantly better on the Head set than on the Tail set at the end of training. This process leads the parameters to converge to $\theta^*$. We showed in theorem 3.3 and 3.8 that under a strongly convex loss function, $\theta^*$ lies within a bounded neighborhood of radius $r$ of the learner's weights $\theta^*_H$ when trained exclusively on the Head set $\mathcal{D}_H$, where $r$ is proportional to the strong convexity of the loss function and inversely proportional to the imbalance factor. This neighborhood falls in $\psi_H$ which represents an area within the weight space where the network performs well on the Head set. At this stage, the model should learn the Tail; however, if it is simply fine-tuned on the Tail ($\mathcal{D}_T$), then it results in moving towards $\theta^*_T$ in $\psi_T$ and will likely leave $\psi_H$.

This phenomenon occurs in sequential learning and it is known as catastrophic forgetting. To mitigate this problem and guide the model to the intersection of $\psi_H$ and $\psi_T$ denoted as $\psi_{HT}$, where the model performs well on both Head and Tail, the Tail should be learned without forgetting the Head. To this end, we propose using the standard CL methods for sequentially learning the Tail after the Head while avoiding catastrophic forgetting (converging to $\theta_{HT}^*$).

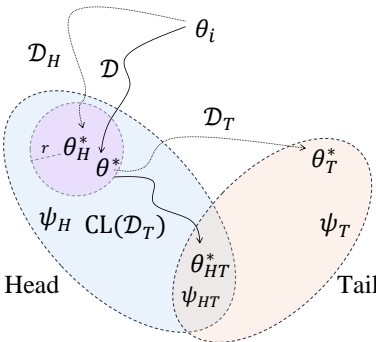

Figure 1: Overview of learning under the LTR scenario and our proposed CLTR approach (symbols described in the text).

Following (Prabhu et al., 2020), a general CL problem can be formulated as a model exposed to a stream of $N$ incoming training datasets $\mathcal{D}_{\mathcal{Y}_t} = \{(x_i, y_i) | y_i \in \mathcal{Y}_t\}$ for $1 \leq t \leq N$, where $\mathcal{Y}_t$ is the corresponding set of labels. Up to the current timestep $t$, the set of labels $\bigcup_{i=1}^{t} \mathcal{Y}_i$ in dataset $\bigcup_{i=1}^{t} \mathcal{D}_{\mathcal{Y}_i}$ has been previously used in training of the network. The objective at the next timestep $t+1$ is to find a mapping $f_\theta : x \to y$ that accurately maps sample $x$ to $\bigcup_{i=1}^{t} \mathcal{Y}_i \cup \mathcal{Y}_{t+1}$, where $\mathcal{Y}_{t+1}$ is the set of new unseen labels in the incoming new dataset $\mathcal{D}_{\mathcal{Y}_{t+1}} = \{(x_i, y_i) | y_i \in \mathcal{Y}_{t+1}\}$. Therefore the ultimate objective of CL is to find an accurate mapping $f_\theta : x \to y$ for all $(x, y) \in \bigcup_{i=1}^{N} \mathcal{D}_{\mathcal{Y}_i}$.

Consider dataset $\mathcal{D}$ under the LTR setup (Definition 3.1) divided into $N$ partitions with substantially different sizes sorted based on cardinality such that $\mathcal{D} = \bigcup_{i=1}^{N} \mathcal{D}^i$ and $|\mathcal{D}^i| >> |\mathcal{D}^{i+1}|$. We have shown in Theorem 3.3 and Theorem 3.8 that $\theta^*$ (the weights of the model when trained on the entire dataset) will be very close to $\theta_1^*$, which is the weights of the model when it is only trained on $\mathcal{D}_1$ (the largest partition of the dataset). As a result, the model after training on $\mathcal{D}$ can be considered as $f_{\theta_1^*} : x \to y$ for all $(x, y) \in \mathcal{D}_1$. On the other hand, following Definition 3.1, the objective of LTR is to learn $f_\theta : x \to y$ for all $(x, y) \in \mathcal{D} = \bigcup_{i=1}^{N} \mathcal{D}^i$. Hence, additional training steps are required for the model to further learn the rest of the partitions of the dataset ($\bigcup_{i=2}^{N} \mathcal{D}^i$). Thus, if we consider each of the partitions of the LTR dataset ($\mathcal{D}^i$ for $1 \leq i \leq N$) as an incoming CL dataset ($\mathcal{D}_{\mathcal{Y}_t}$ for $1 \leq t \leq N$), the objective of the LTR problem would be equivalent to the objective of CL, which is to estimate $f_\theta$:

$$f_\theta : x \to y \quad s.t. \quad (x, y) \in \bigcup_{t=1}^{N} \mathcal{D}_{\mathcal{Y}_t} \quad \text{and} \quad \mathcal{D}_{\mathcal{Y}_1} = \mathcal{D}^1, \ \mathcal{D}_{\mathcal{Y}_2} = \mathcal{D}^2, \ \ldots, \ \mathcal{D}_{\mathcal{Y}_N} = \mathcal{D}^N. \tag{13}$$

Thus, our proposed approach unifies the two domains so that an LTR problem can be treated as a CL problem. Algorithm 1 lays out the detailed procedure of our proposed framework. Due to the higher number of samples in the Head, we start the sequence by learning the Head, a convention also followed in prior multi-stage LTR methods (Zhou et al., 2020; Zhang et al., 2022).

Without loss of generality, we prove the effectiveness of employing a standard and simple CL method for addressing LTR problems in the following theorem. We then extend this notion to other more powerful CL methods empirically in the next section.

**Theorem 3.9.** *Following Assumption 3.2, for the model trained on imbalance dataset $\mathcal{D}$ for i epochs and converging to $\theta^i$, we have*

$$\mathcal{L}(\mathcal{D}, \theta_{CL}^{i+s}) < \mathcal{L}(\mathcal{D}, \theta_{\mathcal{L}}^{i+s}), \tag{14}$$

*for all $s < S - i$, where $s$ is the number of past training epochs in the second phase of training (training on Tail), $S$ is the total number of epochs in both phases of the training, $\theta_{CL}^{i+s}$ and $\theta_{\mathcal{L}}^{i+s}$ denote the weights of the model after $s$ number of updates using CL and regularized cross-entropy loss, respectively.*

*Proof Sketch.* (Formal proof in Appendix A.4) In this theorem, we use the simple EWC loss to represent CL in general. It's important to highlight that while EWC is not the most recent CL technique to be proposed, it serves as a common baseline for comparison of all other CL methods. Furthermore, the mathematical formulation of EWC is succinct and is amenable for use within Theorem 3.9. Moreover, we empirically demonstrate the effectiveness of other and more sophisticated forms of CL for LTR in Section 4.2. We

---

**Algorithm 1** CLTR

---

1: **Input:** imbalanced data $\mathcal{D}$, initialized model parameters $\theta_i$, number of partitions $N$
2: **Output:** $\theta_{HT}^*$
3: sort($\mathcal{D}$) in ascending order by cardinality of each class $|\mathcal{D}_i| \geq |\mathcal{D}_j|$ for all $i < j$
4: partition($\mathcal{D}, L$) where $L = \{l_1, l_2, \ldots, l_N, l_{N+1}\}$ denotes the partition boundaries, $l_1 = 0$, and $l_{N+1} = k$
5: initialize CL replay memory $\mathcal{M}$
6: **for** $i = 1$ **to** $N$ **do**
7: $\quad \mathcal{D}^i = \bigcup_{j=l_i}^{l_{i+1}} \mathcal{D}_j$
8: **end for**
9: **for** $t = 1$ **to** $N$ **do**
10: $\quad \theta_t^* = \arg\min \mathcal{L}_{CL}(\theta, \mathcal{D}^t, \mathcal{M})$ # CL training
11: $\quad$ update $\mathcal{M}$
12: **end for**
13: return $\theta_N^*$

---

consider the updated weights after one iteration using both EWC loss and regularized cross-entropy loss. By employing Taylor expansion, we approximate the losses for the new weights. We then show that the EWC loss incorporates a regularization term that effectively constrains the weight updates. Leveraging the strong convexity of the loss function and the positive nature of the Fisher information matrix, we prove that the loss with EWC-updated weights is strictly less than that with regular cross-entropy updated weights. Note that the loss on the entire dataset $\mathcal{D}$ in Eq. 14 is used as a theoretical upper bound, while this stage of training is solely performed on the Tail set.

## 4 Experiments and Results

### 4.1 Experiment Setup

**Datasets.** First, we use the **MNIST-LT** (LeCun et al., 1998) toy dataset with different IF values and strong convexity parameters to study the behavior of the upper bound (Eq. 10) and its compliance with our theorem. Next, to evaluate the performance of CLTR in addressing the LTR problem, we employ three widely used LTR datasets: **CIFAR100-LT**, **CIFAR10-LT** (Cao et al., 2019), and **ImageNet-LT** (Liu et al., 2019). These datasets represent long-tailed versions of the original CIFAR100, CIFAR10, and ImageNet datasets, maintaining the same number of classes while the number of samples in each class decreases exponentially according to the *IF*, where the first class has the maximum number of samples and the last class contains the least number of samples, as illustrated in Appendix D. Finally, to further highlight the benefits of using CLTR, we carry out additional experiments using the naturally skewed **Caltech256** dataset (Griffin et al., 2007).

**Implementation Details.** Following the experimental setup of (Alshammari et al., 2022) and (Fu et al., 2022), we use ResNet-32 (He et al., 2016) and ResNeXt-50 (Xie et al., 2017) for CIFAR and ImageNet benchmarks, respectively. The LTR methods selected for comparison are state-of-the-art solutions in the area. We also employ various standard and state-of-the-art CL method in CLTR, namely LwF (Li & Hoiem, 2017), EWC (Kirkpatrick et al., 2017), Modified EWC (Molahasani et al., 2023), GPM (Saha et al., 2020), FOSTER (Wang et al., 2022), SGP (Saha & Roy, 2023), and TPL (Lin et al., 2024a). We divided the dataset into 2 partitions for LwF, EWC, Modified EWC, GPM, and SGP and 4 partitions for FOSTER and TPL. All trainings were conducted using an NVIDIA RTX 3090 GPU with 24GB VRAM. More details on the implementation specifics are provided in Appendix C.

**Evaluation.** For the LTR datasets (MNIST-LT, CIFAR100-LT, CIFAR10-LT, ImageNet-LT), we first train the model on the long-tailed imbalanced training set and then evaluate it on the balanced test set, following the evaluation protocol of (Alshammari et al., 2022). For Caltech256, we use the entire training set for training and assess the model's performance on the entire test set, retaining its original distribution. All

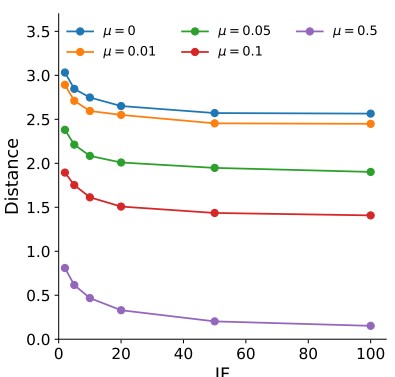

Figure 2: The distance between $\theta^*$ and $\theta_H^*$ in different IF and $\mu$.

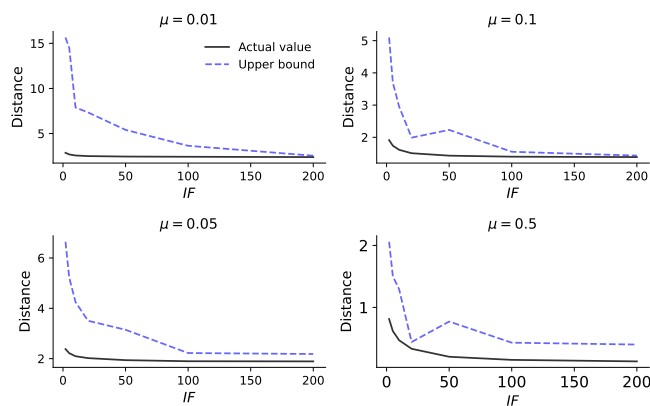

Figure 3: The actual distance between $\theta^*$ and $\theta_H^*$ in different IF and $\mu$ compared with the calculated upper bound.

reported values represent classification accuracy. The results of our proposed approach are  highlighted  in the Tables.

### 4.2 Results

**Empirical support for Theorem 3.3.** To evaluate the validity of Theorem 3.3 on the upper bound for the distance between the learner's weights when trained on $\mathcal{D}$ and $\mathcal{D}_H$ ($\|\theta^* - \theta_H^*\|$), we first train a logistic regression model on MNIST-LT with varying IF and $\mu$ values. Then we calculate the Euclidean distance between the two sets of weights, as illustrated in Fig. 2. As expected from Eq. 10, increasing either the IF or strong convexity ($\mu$) results in a reduced distance, indicating that the weights of the model trained using $\mathcal{D}$ approach the weights when it is solely trained using $\mathcal{D}_H$. We also compared these actual distances with the upper bound predicted by Theorem 3.3, as exhibited in Fig. 3. The results show that for all IF and $\mu$ values, the measured distance is lower than the theoretical upper bound, which is aligned with our proposed theorem. It is important to note that for this experiment, the upper bound is calculated using Eq. 5 in Appendix A.1 which results in even a tighter neighborhood compared to Eq. 10.

**Performance.** We compare the performance of our CLTR framework with existing state-of-the-art LTR solutions on three LTR benchmarks, CIFAR100-LT, CIFAR10-LT, and ImageNet-LT, as presented in Tables 1, 2, and 3. We also present two additional baselines where we train the backbone model on the imbalanced data, with and without a class-balanced loss term. These results demonstrate that CLTR indeed provides the best or the second-to-best performance across all benchmarks, as predicted by our proposed theorems.

Following the prior works such as (Alshammari et al., 2022), we avoid direct comparisons with solutions with "bells and whistles" such as RIDE (Wang et al., 2021c), ACE (Cai et al., 2021), SSD (Li et al., 2021), and PaCo (Cui et al., 2021), which employ aggressive data augmentations, ensembles learning, multi-expert and self-supervised pretraining. It is worth mentioning that some previous LTR solutions like BBN (Zhou et al., 2020) learn the Head and Tail separately in a multi-stage manner. They rely on various techniques to prevent performance loss on the Head while learning the Tail. However, unlike these methods, our approach only uses one model through the entire training process, and the results demonstrate that employing standard CL methods designed to mitigate catastrophic forgetting yields the best performance in the LTR benchmarks.

To further explore the capabilities of our approach in more challenging settings, we compare the performance of CLTR in addressing the LT-CIL problem with the prior state-of-the-art solutions in the area. Following the experimental setup in (Liu et al., 2022a; Hou et al., 2019; Douillard et al., 2020), the models are first trained on the largest 50 classes (Head), then, the remaining classes are learned incrementally in 5 or 10 consecutive tasks (Tail) with an equal number of new classes in each new task, from the largest subset to the smallest subset of the dataset. We apply our method in this setting on the CIFAR100-LT dataset and compare its performance with the prior works, as presented in Table 4. The results demonstrate that

Table 1: LTR benchmarks for CIFAR100-LT.

| Model | IF | | |
|---|---|---|---|
| | 100 | 50 | 10 |
| Baseline (Cui et al., 2019) | 38.3 | 43.9 | 55.7 |
| Baseline + CB (Cui et al., 2019) | 39.6 | 45.3 | 58.0 |
| Focal loss (Lin et al., 2017) | 38.4 | 44.3 | 55.8 |
| Focal+CB (Cui et al., 2019) | 39.6 | 45.2 | 58.0 |
| $\tau$-norm (Kang et al., 2019) | 47.7 | 52.5 | 63.8 |
| LDAM-DRW (Cao et al., 2019) | 42.0 | 46.6 | 58.7 |
| BBN* (Zhou et al., 2020) | 42.6 | 47.0 | 59.1 |
| LogitAjust (Menon et al., 2020) | 42.0 | 47.0 | 57.7 |
| LDAM+SSP (Yang & Xu, 2020) | 43.4 | 47.1 | 58.9 |
| De-confound (Tang et al., 2020) | 44.1 | 50.3 | 59.6 |
| SSD (Li et al., 2021) | 46.0 | 50.5 | 62.3 |
| DiVE (He et al., 2021) | 45.4 | 51.1 | 62.0 |
| DRO-LT (Samuel & Chechik, 2021) | 47.3 | 57.6 | 63.4 |
| WD (Alshammari et al., 2022) | 46.0 | 52.7 | 66.1 |
| WD & Max (Alshammari et al., 2022) | **53.4** | 57.7 | **68.7** |
| CLTR (LwF) | 45.1 | 49.3 | 58.7 |
| CLTR (EWC) | 44.4 | 50.3 | 58.8 |
| CLTR (Modified EWC) | 45.9 | 51.0 | 60.7 |
| CLTR (GPM) | 48.3 | 54.7 | 64.7 |
| CLTR (FOSTER) | 48.7 | 54.4 | 63.6 |
| CLTR (SGP) | 50.7 | **58.0** | 67.2 |
| CLTR (TPL) | 48.4 | 54.0 | 62.1 |

Table 2: LTR benchmarks for CIFAR10-LT.

| Model | IF | |
|---|---|---|
| | 100 | 50 |
| Baseline (Cui et al., 2019) | 69.8 | 75.2 |
| Baseline + CB (Cui et al., 2019) | 74.7 | 79.3 |
| Focal loss(Lin et al., 2017) | 70.4 | 75.3 |
| PG Re-sampling (Cui et al., 2018) | 67.1 | 75.0 |
| 3LSSL (Díaz-Rodríguez et al., 2018) | 85.2 | 88.2 |
| Focal+CB(Cui et al., 2019) | 74.6 | 79.3 |
| LDAM-DRW(Cao et al., 2019) | 77.0 | 79.3 |
| BBN* (Cao et al., 2019) | 79.8 | 82.2 |
| Manifold mixup (Cui et al., 2019) | 73.0 | 78.1 |
| CBA-LDAM (Cui et al., 2019) | 80.3 | 82.2 |
| ELF (LDAM)+DRW (Cui et al., 2019) | 78.1 | 82.4 |
| De-confound (Tang et al., 2020) | 80.6 | 83.6 |
| Hybrid-SC (Wang et al., 2021b) | 81.4 | 85.4 |
| MiSLAS (Zhong et al., 2021) | 82.1 | 85.7 |
| BCL (Zhu et al., 2022) | 84.3 | 87.2 |
| CLTR (LwF) | 76.3 | 78.6 |
| CLTR (EWC) | 75.1 | 80.1 |
| CLTR (Modified EWC) | 77.8 | 81.3 |
| CLTR (GPM) | 81.2 | 84.8 |
| CLTR (FOSTER) | 81.7 | 85.9 |
| CLTR (SGP) | 83.0 | 85.5 |
| CLTR (TPL) | **84.7** | **87.6** |

Table 3: LTR benchmarks for ImageNet-LT.

| Model | Top-1 accuracy |
|---|---|
| Baseline (Cui et al., 2019) | 44.4 |
| Baseline + CB (Cui et al., 2019) | 33.2 |
| KD (Hinton et al., 2015) | 35.8 |
| Focal (Lin et al., 2017) | 30.5 |
| SR Re-sampling (Mahajan et al., 2018) | 46.8 |
| OLTR (Liu et al., 2018) | 35.6 |
| cRT (Kang et al., 2019) | 49.6 |
| $\tau$-norm (Kang et al., 2019) | 49.4 |
| LFME (Xiang et al., 2020) | 37.5 |
| De-confound (Tang et al., 2020) | 51.8 |
| Seasaw Loss (Wang et al., 2021a) | 50.4 |
| DiVE (He et al., 2021) | 53.1 |
| DisAlign (Zhang et al., 2021) | 52.9 |
| WD (Alshammari et al., 2022) | 48.6 |
| WD+Max (Alshammari et al., 2022) | **53.9** |
| CLTR (LwF) | 47.6 |
| CLTR (EWC) | 48.9 |
| CLTR (Modified EWC) | 49.1 |
| CLTR (GPM) | 51.7 |
| CLTR (FOSTER) | 52.7 |
| CLTR (SGP) | 53.2 |
| CLTR (TPL) | **53.9** |

Table 4: The performance of CLTR on Ordered LT-CIL Benchmark for CIFAR100-LT .

| Method | Tasks | |
|---|---|---|
| | 5 | 10 |
| EEIL (Castro et al., 2018) | 38.5 | 37.5 |
| EEIL+2sLWS Liu et al. (2022a) | 39.0 | 37.6 |
| LUCIR (Hou et al., 2019) | 42.7 | 42.2 |
| PODNET (Douillard et al., 2020) | 44.1 | 44.0 |
| PODNET+2sLWS (Liu et al., 2022a) | 44.4 | 44.4 |
| LUCIR+2sLWS (Liu et al., 2022a) | 45.9 | 45.7 |
| CLTR (TPL) | **48.4** | **47.3** |

Table 5: The performance of CLTR on Caltech256.

| Method | Backbone | |
|---|---|---|
| | Inc.V4 | Res.101 |
| $L^2 - FE$ (Li et al., 2018) | 84.1 | 85.3 |
| $L^2$ (Li et al., 2018) | 85.8 | 87.2 |
| $L^2 - SP$ (Li et al., 2018) | 85.3 | 87.2 |
| DELTA (Li et al., 2018) | 86.8 | 88.7 |
| GBN (Liu et al., 2021) | - | 86.9 |
| TransTailor (Liu et al., 2021) | - | 87.3 |
| CLTR (SGP) | **88.6** | **89.8** |

CLTR outperforms prior methods in both 5- and 10-task settings by considerable margins of 2.5% and 1.6%, repsectively.

In LTR benchmarks, datasets are modified to exhibit a skewed distribution of samples among various classes. However, such imbalanced class distributions are naturally observed in real-world data as well (Alshammari et al., 2022). To evaluate the efficacy of CL techniques on non-LTR benchmark datasets, we utilize the Caltech256 dataset (Griffin et al., 2007), which consists of 256 distinct classes representing everyday objects. The largest class comprises 827 samples, while the smallest class contains only 80 samples, exhibiting an *IF* of over 10. Here, we employ the CLTR and compare its performance to the state-of-the-art methods on this

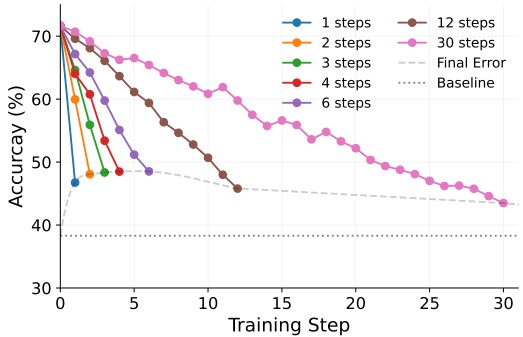 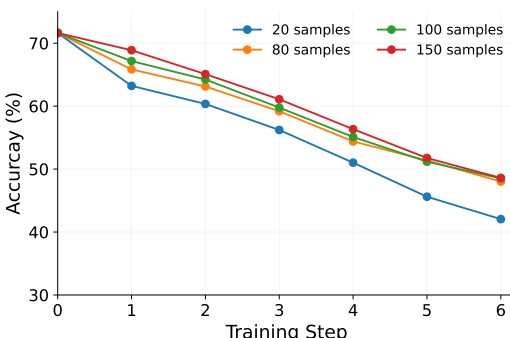

Figure 4: The Error behavior for various (left) number of incremental steps and (right) replay memory ($\mathcal{M}$) size.

dataset for objected classification. The results are presented in Table 5. We observe that CL outperforms the previous method on this dataset, demonstrating the strong potential of using CL in dealing with long-tailed real-world datasets.

### 4.3 Discussion

**Multiple Incremental Steps.** Recall that Theorem 3.8 highlights CLTR's capability for extending beyond two incremental steps. Increasing the number of partitions leads to smaller IF within each partition at the cost of an increase in forgetting. To explore the effect of varying partition numbers on CLTR's final performance, we adhere to the experimental protocol outlined in (Liu et al., 2022a). Initially, the model is trained on the first 60 classes, followed by sequential learning of the remaining classes, divided into different numbers of partitions. The results of this experiment are presented in Fig. 4 (left). Our results reveal an optimal value for CLTR (FOSTER) (4 steps), yet the performance margin remains slim even with up to 30 tasks. This highlights the effectiveness of CL methods employed within CLTR. The optimal value for the number of incremental steps for each CL algorithm can be found in Appendix C.

**Replay Memory.** Several CL algorithms incorporate mechanisms to retain partial information from the previous task, aiming to mitigate catastrophic forgetting. For example, EWC maintains prior model parameters along with their Fisher values, whereas both GPM and SGP safeguard the Core Gradient Space of previous tasks. FOSTER, on the other hand, utilizes a replay memory for this purpose. Within the LTR context, the presence of a buffer memory doesn't require additional storage, as access to the full dataset is already available. Nonetheless, to prevent hindering the model's capacity to learn Tail distributions, we deliberately avoid replaying all Head samples when learning the Tail, as evidenced by Eq. 7. Accordingly, our analysis extends to how the number of Head samples replayed while learning the Tail impacts the model's performance, as illustrated in Fig. 4 (right). The replay memory's size serves as a mediator between forgetting previous information and worsening class imbalance, e.g. a larger replay memory reduces forgetting but increases imbalance. Therefore, identifying an optimal balance in this trade-off is crucial. Our results demonstrate the significance of an appropriate replay memory size; however, there exists a threshold beyond which additional samples per class do not further improve the performance and the performance levels off.

**Backward/Forward Transfer and Catastrophic Forgetting.** Prior works discuss three key concepts in the context of CL: catastrophic forgetting, backward transfer, and forward transfer (Díaz-Rodríguez et al., 2018). As mentioned earlier, catastrophic forgetting occurs when the performance of a class declines after retraining. Despite the use of CL methods, which are designed to mitigate this forgetting, a certain degree of forgetting is still inevitable. Forward transfer is the improvement in performance on a new task after employing CL, which is the central aim of retraining in CL. Finally, backward transfer is a beneficial side-effect where retraining on new samples can actually enhance the model's performance on the previous tasks. This interesting phenomenon in CL has been extensively discussed in previous works in theory and practice (Lin et al., 2022). Now, let's discuss Fig. 5, which presents the difference in per-class accuracy of the best

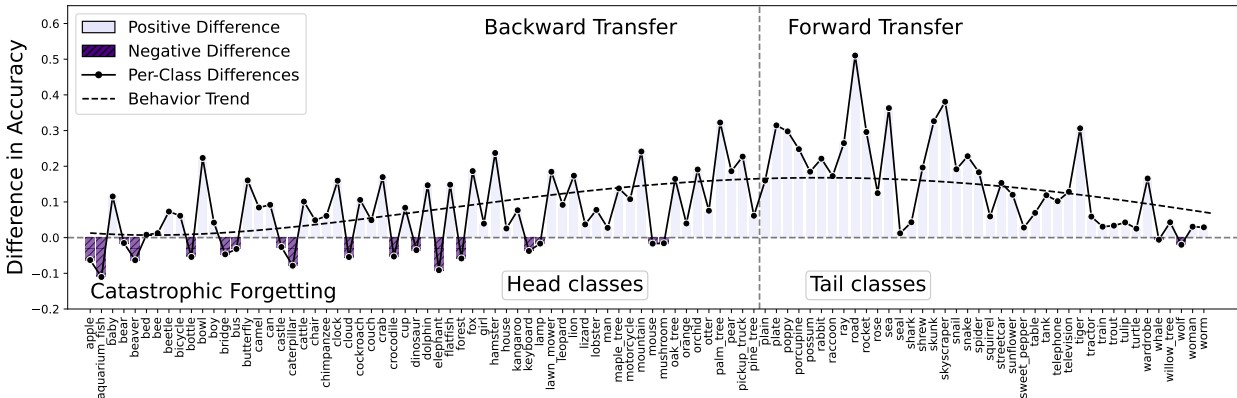

Figure 5: The difference in per-class accuracy of CLTR (SGP) and the baseline model. 🔍

CL method (CLTR (SGP)) versus the baseline network. The analysis is based on CIFAR100-LT with an *IF* of 100. The figure is divided into three regions corresponding to the scenarios discussed above: catastrophic forgetting (bottom), backward transfer (top-left), and forward transfer (top-right). The bottom region in the figure represents classes that undergo catastrophic forgetting, while the top-right region represents the Tail samples (with a class index larger than 60), which demonstrate improved performance, or forward transfer. We observe that using SGP as a CL solution for LTR results in very effective improvements in the per-class accuracy of the Tail (forward transfer). Interestingly, despite the absence of Head data in the retraining process, 42 out of 60 Head classes see some level of improvement after the model is exposed to the Tail samples (backward transfer). This result emphasizes the remarkable potential of CL methods in enhancing the performance on both new and previous tasks.

**Runtime.** The inference runtime is identical between CLTR and LTR solutions, due to identical backbones in both types of methods and the fact that CL does not affect inference. Regarding the training runtime, when CLTR is used, the data is divided into Head and Tail sets. At each step of the training, only one partition of data is involved, alongside a replay memory with a limited size. Since the backbone is consistent among all LTR approaches for each benchmark, the runtime is determined by the amount of data fed to the model. Dividing the learning into multiple steps and using CL therefore does not impact the total runtime, nor does it increase the training time significantly.

**Limitations.** Strong convexity is a key assumption in our theorem, which determines an upper bound for the distance between the weights of a learner trained on the full dataset and the weights of the same learner trained solely on the Head. This assumption offers a solid theoretical foundation for our method, showcasing the feasibility of using CL techniques to address the LTR problem. However, as many deep learning models in practice employ non-convex loss functions that potentially limit the theorem's applicability to specific cases, it is crucial to highlight that our experimental results are not strictly dependent on the strong convexity condition. In fact, our method exhibits impressive performance even under more relaxed conditions, indicating its robustness and adaptability.

## 5 Conclusion and Future Work

In this work, we propose CLTR, a novel framework that uses standard CL techniques to learn the Head and Tail sets sequentially. To ensure that our proposed solution is theoretically grounded, we first prove that learning a long-tailed dataset leads to weights similar to the case where the model is solely trained on the Head. Relying on this finding, we propose CL for learning the Tail sequentially following the Head, without forgetting the Head. Our experimental results on CIFAR100-LT, CIFAR10-LT, ImageNet-LTR, and Clatech256 support our theoretical findings and demonstrate the viability of our approach in achieving state-of-the-art performances in all LTR and LT-CIL benchmarks. Future research directions include relaxing some of our theoretical assumptions, and employing few-shot learning alongside CL for addressing LTR.

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

## Appendix

## A  Proofs

### A.1  Proof of Lemma 3.4

*Proof.* Since $f(x)$ is strongly convex:

$$f(x_2) \geq f(x_1) + \nabla f(x_1)^T(x_2 - x_1) + \frac{\mu_f}{2}\|x_2 - x_1\|^2. \tag{1}$$

Accordingly if $x_2 = x_g = \arg\min g(x)$ and $x_1 = x_f = \arg\min f(x)$, then:

$$f(x_g) - f(x_f) \geq \nabla f(x_f)^T(x_g - x_f) + \frac{\mu_f}{2}\|x_g - x_f\|^2. \tag{2}$$

Since $x_f$ is the minimizer of $f$, $\nabla f(x_f) = 0$. Therefore:

$$f(x_g) - f(x_f) \geq \frac{\mu_f}{2}\|x_g - x_f\|^2. \tag{3}$$

Similarly, considering $g(x)$, with $x_1 = x_g$, and $x_2 = x_f$, we can derive Equation 1 as follows:

$$g(x_f) - g(x_g) \geq \frac{\mu_g}{2}\|x_f - x_g\|^2. \tag{4}$$

By adding and rearranging Eqs. 3 and 4, we will have:

$$(g(x_f) - f(x_f)) + (f(x_g) - g(x_g)) \geq \frac{(\mu_f + \mu_g)}{2}\|x_g - x_f\|^2. \tag{5}$$

Using $|f(x) - g(x)| \leq \delta$, we can maximize $(g(x_f) - f(x_f))$ and $(f(x_g) - g(x_g))$ to obtain:

$$2\delta \geq \frac{\mu_f + \mu_g}{2}\|x_g - x_f\|^2. \tag{6}$$

Hence:

$$\|x_g - x_f\|^2 \leq \frac{4\delta}{\mu_f + \mu_g}, \tag{7}$$

which completes the proof. $\square$

### A.2  Proof of Lemma 3.6

*Proof.* Using the second-order Taylor series expansion for multivariate functions, we can approximate $f(x_g)$ and $g(x_f)$ as follows:

$$f(x_g) \simeq f(x_f) + \nabla f(x_f)(x_g - x_f) + \frac{1}{2}(x_g - x_f)^\top H_f(x_f)(x_g - x_f), \tag{8}$$

$$g(x_f) \simeq g(x_g) + \nabla g(x_g)(x_f - x_g) + \frac{1}{2}(x_f - x_g)^\top H_g(x_g)(x_f - x_g), \tag{9}$$

where $H_f(x_f)$ and $H_g(x_g)$ are the Hessian matrices of $f$ and $g$ evaluated at $x_f$ and $x_g$, respectively.

Since $\nabla f(x_f) = \nabla g(x_g) = 0$, by adding Eq. 8 and Eq. 9 together, we obtain:

$$f(x_g) - g(x_g) + g(x_f) - f(x_f) \simeq \frac{1}{2}(x_g - x_f)^\top H_f(x_f)(x_g - x_f) + \frac{1}{2}(x_f - x_g)^\top H_g(x_g)(x_f - x_g), \tag{10}$$

Using $|f(x) - g(x)| \leq \delta$, we can maximize $(g(x_f) - f(x_f))$ and $(f(x_g) - g(x_g))$:

$$2\delta \geq \frac{1}{2}(x_g - x_f)^\top H_f(x_f)(x_g - x_f) + \frac{1}{2}(x_f - x_g)^\top H_g(x_g)(x_f - x_g), \tag{11}$$

Let $\lambda_f$ and $\lambda_g$ be the minimum eigenvalues of $H_f(x_f)$ and $H_g(x_g)$, respectively. By properties of the minimum eigenvalues, we can say:

$$(x_g - x_f)^\top H_f(x_f)(x_g - x_f) \geq \lambda_f \|x_g - x_f\|^2, \tag{12}$$

$$(x_f - x_g)^\top H_g(x_g)(x_f - x_g) \geq \lambda_g \|x_f - x_g\|^2. \tag{13}$$

Using Eqs. 12 and 13, we can rewrite Eq. 11:

$$2\delta \geq \frac{1}{2}\lambda_f \|x_g - x_f\|^2 + \frac{1}{2}\lambda_g \|x_f - x_g\|^2. \tag{14}$$

Therefore:

$$\|x_f - x_g\|^2 \leq \frac{4\delta}{\lambda_f + \lambda_g}, \tag{15}$$

which completes the proof. $\qquad\square$

## A.3 Proof of Theorem 3.8

*Proof.* Let $\mathcal{D}$ be a dataset divided into a sequence of partitions $\mathcal{D}^1, \mathcal{D}^2, \ldots, \mathcal{D}^n$ such that the imbalance factor between any two consecutive partitions $\mathcal{D}^i$ and $\mathcal{D}^{i+1}$ is significantly large, i.e., $\frac{|\mathcal{D}^i|}{|\mathcal{D}^{i+1}|} \gg 1$.

Consider a random subset of $\mathcal{D}$ sorted from largest to smallest denoted as $\mathcal{D}^a, \mathcal{D}^b, \mathcal{D}^c, \ldots$ (where $|\mathcal{D}^a| \gg |\mathcal{D}^b| \gg |\mathcal{D}^c|$).

From Theorem 3.3, we know that if the imbalance factor between two partitions is significantly large, $\frac{|\mathcal{D}^1|}{|\mathcal{D}^2|} \gg 1$, then the distance between the optimal parameters when trained on $\mathcal{D}^1$ and $\mathcal{D}^1 \cup \mathcal{D}^2$ is bounded by $\zeta$, i.e., $\|\theta_{\mathcal{D}^1}^* - \theta_{\mathcal{D}^1 \cup \mathcal{D}^2}^*\|^2 \leq \zeta$ where $\zeta$ is computed using Eq. 9 in the manuscript.

Applying this Theorem to $\mathcal{D}^a$ and $\mathcal{D}^b$, we have:

$$\|\theta_{\mathcal{D}^a}^* - \theta_{\mathcal{D}^a \cup \mathcal{D}^b}^*\|^2 \leq \zeta_1$$

Next, considering the combination of $\mathcal{D}^a \cup \mathcal{D}^b$ and $\mathcal{D}^c$, given that $\frac{|\mathcal{D}^a \cup \mathcal{D}^b|}{|\mathcal{D}^c|} \gg 1$, we deduce:

$$\|\theta_{\mathcal{D}^a \cup \mathcal{D}^b}^* - \theta_{\mathcal{D}^a \cup \mathcal{D}^b \cup \mathcal{D}^c}^*\|^2 \leq \zeta_2$$

Given that the weights reside in a metric space, and the distances are Euclidean, the triangle inequality applies. Combining the above inequalities, we therefore get:

$$\|\theta_{\mathcal{D}^a}^* - \theta_{\mathcal{D}^a \cup \mathcal{D}^b \cup \mathcal{D}^c}^*\|^2 \leq (\sqrt{\zeta_1} + \sqrt{\zeta_2})^2$$

Extending this argument for all partitions, we can conclude:

$$\|\theta_{\mathcal{D}^a}^* - \theta_{\sum \mathcal{D}^i}^*\|^2 \leq \left(\sum_{i=1}^m \sqrt{\zeta_i}\right)^2$$

where $m$ is the number of subsets selected randomly. $\qquad\square$

## A.4 Proof of Theorem 3.9

*Proof.* Define the updated weight vector after one iteration over the Tail using EWC loss as:

$$\theta_{\text{EWC}}^{i+s} = \theta^{i+s-1} - \eta \nabla \mathcal{L}_{\text{EWC}}(\mathcal{D}_T, \theta^{i+s-1}) \tag{16}$$

Similarly, for $\mathcal{L}$:

$$\theta_{\mathcal{L}}^{i+s} = \theta^{i+s-1} - \eta \nabla \mathcal{L}(\mathcal{D}_T, \theta^{i+s-1}) \tag{17}$$

From the Taylor series expansion, we can estimate the $\mathcal{L}$ of the model with $\theta_{\text{EWC}}^{i+s}$ over $\mathcal{D}$:

$$\mathcal{L}(\mathcal{D}, \theta_{\text{EWC}}^{i+s}) \simeq \mathcal{L}(\mathcal{D}, \theta^{i+s-1}) - \eta \nabla \mathcal{L}_{\text{EWC}}(\mathcal{D}_T, \theta^{i+s-1}) \nabla \mathcal{L}(\mathcal{D}, \theta^{i+s-1}) \tag{18}$$

Similarly, for the $\mathcal{L}$ of the model with $\theta_{\mathcal{L}}^{i+s}$ over $\mathcal{D}$:

$$\mathcal{L}(\mathcal{D}, \theta_{\mathcal{L}}^{i+s-1}) \simeq \mathcal{L}(\mathcal{D}, \theta^{i+s-1}) - \eta \nabla \mathcal{L}(\mathcal{D}_T, \theta^{i+s-1}) \nabla \mathcal{L}(\mathcal{D}, \theta^{i+s-1}) \tag{19}$$

Subtracting Eq. 19 from 18, we derive:

$$\mathcal{L}(\mathcal{D}, \theta_{\text{EWC}}^{i+s}) - \mathcal{L}(\mathcal{D}, \theta_{\mathcal{L}}^{i+s}) \simeq \eta \nabla \mathcal{L}(\mathcal{D}, \theta^{i+s-1}) (\nabla \mathcal{L}(\mathcal{D}_T, \theta^{i+s-1}) - \nabla \mathcal{L}_{\text{EWC}}(\mathcal{D}_T, \theta^{i+s-1})) \tag{20}$$

Elastic Weight Consolidation (EWC) loss is expressed as:

$$\mathcal{L}_{\text{EWC}}(\theta^{i+s-1}) = \mathcal{L}(\theta^{i+s-1}) + \frac{\lambda}{2} \sum_i^{|\theta|} F_i (\theta^{i+s-1} - \theta^*)^2 \tag{21}$$

Thus, we can compute $\nabla \mathcal{L}_{\text{EWC}}(\mathcal{D}_T, \theta^{i+s-1})$ as:

$$\nabla \mathcal{L}_{\text{EWC}}(\mathcal{D}_T, \theta^{i+s-1}) = \nabla \mathcal{L}(\mathcal{D}_T, \theta^{i+s-1}) + \lambda \text{diag}(F)(\theta^{i+s-1} - \theta^*) \tag{22}$$

Substituting Eq. 22 into Eq. 20, we obtain:

$$\mathcal{L}(\mathcal{D}, \theta_{\text{EWC}}^{i+s}) - \mathcal{L}(\mathcal{D}, \theta_{\mathcal{L}}^{i+s}) = -\eta \lambda \text{diag}(F) \nabla \mathcal{L}(\mathcal{D}, \theta^{i+s-1})^T (\theta^{i+s-1} - \theta^*) \tag{23}$$

To determine the sign of $\eta \lambda \text{diag}(F) \nabla \mathcal{L}(\mathcal{D}, \theta^{i+s-1})^T (\theta^{i+s-1} - \theta^*)$, we must investigate the sign of each factor. The values of $\eta$ and $\lambda$ are positive by construction. To determine the sign of $\nabla \mathcal{L}(\mathcal{D}, \theta^{i+s-1})^T (\theta^{i+s-1} - \theta^*)$, based on the strong convexity of $\mathcal{L}$ with respect to $\theta^i$ and $\theta^*$, we have:

$$\mathcal{L}(\mathcal{D}, \theta^*) \geq \mathcal{L}(\mathcal{D}, \theta^{i+s-1}) + \nabla \mathcal{L}(\mathcal{D}, \theta^{i+s-1})^T (\theta^* - \theta^{i+s-1}) + \frac{\mu_{\mathcal{L}}}{2} |\theta^{i+s-1} - \theta^*|^2. \tag{24}$$

Rearranging, we obtain:

$$\nabla \mathcal{L}(\mathcal{D}, \theta^{i+s-1})^T (\theta^* - \theta^{i+s-1}) \leq \mathcal{L}(\mathcal{D}, \theta^*) - \mathcal{L}(\mathcal{D}, \theta^{i+s-1}) - \frac{\mu_{\mathcal{L}}}{2} \|\theta^{i+s-1} - \theta^*\|^2. \tag{25}$$

Since $\theta^*$ minimizes $\mathcal{L}$, the term $\mathcal{L}(\mathcal{D}, \theta^*) - \mathcal{L}(\mathcal{D}, \theta^{i+s-1})$ is always negative. Moreover, $-\frac{\mu_{\mathcal{L}}}{2} \|\theta^i - \theta^*\|^2$ is also always negative, leading to:

$$\nabla \mathcal{L}(\mathcal{D}, \theta^{i+s-1})^T (\theta^* - \theta^{i+s-1}) < 0. \tag{26}$$

Consequently, $\nabla \mathcal{L}(\mathcal{D}, \theta^{i+s-1})^T (\theta^i - \theta^*)$ is positive definite.

Finally, the $\text{diag}(F)$ term is determined to be positive valued, according to the following Lemma A.1.

**Lemma A.1.** *Let a logistic regression model be characterized by parameters $\theta$ and trained using regularized cross-entropy loss. Then, the diagonal values of its Fisher information matrix ($\text{diag}(F)$) are strictly positive.*

The full proof is presented in A.5. Relying on Lemma A.1, we have derived that the sign of $\eta \lambda \text{diag}(F) \nabla \mathcal{L}(\mathcal{D}, \theta^{i+s-1})^T (\theta^i - \theta^*)$ is positive, which from Eq. 23 we can conclude:

$$\mathcal{L}(\mathcal{D}, \theta_{EWC}^{i+s}) - \mathcal{L}(\mathcal{D}, \theta_{\mathcal{L}}^{i+s}) < 0, \tag{27}$$

which completes the proof. □

### A.5 Proof of Lemma A.1

*Proof.* The Fisher information matrix is the estimated value of the Hessian of the log-likelihood:

$$F = \mathbb{E}\left[\nabla^2(-\log \mathcal{L}(\theta))\right] \tag{28}$$

In logistic regression, we model the probability of a binary outcome $y$ given input $\mathbf{x}$ as:

$$P(y = 1|\mathbf{x}; \theta) = \frac{1}{1 + e^{-\theta^T \mathbf{x}}} \tag{29}$$

where $\theta$ is the vector of model parameters. For a dataset $\{(\mathbf{x}_i, y_i)\}_{i=1}^N\}$, the negative log-likelihood is:

$$-\log \mathcal{L}(\theta) = \sum_{i=1}^N \left[-y_i \log\left(\frac{1}{1 + e^{-\theta^T \mathbf{x}_i}}\right) - (1 - y_i) \log\left(1 - \frac{1}{1 + e^{-\theta^T \mathbf{x}_i}}\right)\right] \tag{30}$$

So the Hessian of the negative log-likelihood is:

$$\nabla^2(-\log \mathcal{L}(\theta)) = \begin{bmatrix} \frac{\partial^2(-\log \mathcal{L})}{\partial \theta_1^2} & \cdots & \frac{\partial^2(-\log \mathcal{L})}{\partial \theta_1 \partial \theta_d} \\ \vdots & \ddots & \vdots \\ \frac{\partial^2(-\log \mathcal{L})}{\partial \theta_d \partial \theta_1} & \cdots & \frac{\partial^2(-\log \mathcal{L})}{\partial \theta_d^2} \end{bmatrix} \tag{31}$$

As a result:

$$\nabla^2(-\log \mathcal{L}(\theta)) = \nabla^2 L(\theta) \tag{32}$$

where $d$ is the dimensionality of $\theta$. Now since the model is logistic regression and loss is regularized cross-entropy, from Eq. 9, we have:

$$\mathcal{L}(x_1) \geq \mathcal{L}(x_2) + \nabla \mathcal{L}(x_2)^T(x_1 - x_2) + \frac{\mu_{\mathcal{L}}}{2}\|x_1 - x_2\|^2, \tag{33}$$

Which is the condition of strong convexity. As a result:

$$\nabla^2 \mathcal{L} \geq \mu_{\mathcal{L}} \boldsymbol{I} \tag{34}$$

From Eq.32 and Eq. 34:

$$\nabla^2(-\log \mathcal{L}(\theta)) = \nabla^2 L(\theta) \geq \mu I \tag{35}$$

Hence:

$$\mathbb{E}\left[\nabla^2(-\log \mathcal{L}(\theta))\right] \geq \mu I \tag{36}$$

consequently:

$$\text{diag}(F) > \text{diag}(D), \quad \text{where } D_{ii} > 0, \quad \text{for all } i \tag{37}$$

which completes the proof. □

## B  Strong Convexity of the Loss Function

The assumption of strong convexity has been widely used in prior theoretical analysis of CL algorithms (Lin et al., 2024b; Wu et al., 2024; Hao et al., 2024; Cai & Diakonikolas, 2024; Zeno et al., 2021; Bennani et al., 2020). Here we aim to investigate whether the sequence in which training on multiple datasets is conducted affects the outcome when the loss abides by this assumption. We begin by illustrating how the loss landscape and its minima are influenced by the training data. We then establish that when training the model on multiple datasets, altering the sequence of training impacts the convergence point under the strong convexity condition. This is then followed by simple numerical examples that empirically demonstrate how different training orders yield different results.

Following (Picot et al., 2022), we define the empirical loss landscape $\mathcal{E}$ as the change of the loss function with respect to the change in the parameters of the model ($\theta$) when training on a particular dataset $\mathcal{D}_i$. Therefore the loss landscape can be formulated as:

$$\mathcal{E}_{\mathcal{D}_i}(\theta) = \mathcal{L}(\theta, \mathcal{D})|_{\mathcal{D}=\mathcal{D}_i} = \frac{1}{|\mathcal{D}|} \sum_{k=1}^{|\mathcal{D}|} \ell((x_k, y_k), \theta)\bigg|_{(x_k, y_k) \in \mathcal{D}_i}. \tag{1}$$

Hence, the parameters of the model after training can be expressed as:

$$\theta_i^* = \arg\min_\theta \mathcal{E}_{\mathcal{D}_i}(\theta) = \arg\min_\theta \frac{1}{|\mathcal{D}|} \sum_{k=1}^{|\mathcal{D}|} \ell((x_k, y_k), \theta)\bigg|_{(x_k, y_k) \in \mathcal{D}_i}. \tag{2}$$

Since the loss function is assumed to be strongly convex, there will be only one minima (the global minima), and the convergence of SGD to this point, under proper selection of the learning rate, is guaranteed (Rakhlin et al., 2011). As a result, regardless of the initialization, for each dataset, the model will converge to its corresponding global minima in the weight space. Consequently, if the model is trained on a sequence of different datasets, it will always converge to the global minimum corresponding to the final dataset. So the order of training can significantly change the outcome. We further demonstrate this phenomenon in the following simple numerical examples.

We consider a simple dataset $\mathcal{D}_1$ consisting of two data points $\{(0, 0), (1, 1)\}$ with labels $\{0, 1\}$. We train a logistic regression model with L2 regularization (following Assumption 3.2) on this dataset which falls under the strong convexity assumption, following Eq. 8. We use randomly initialized parameters $\theta_{init}$ and compare the corresponding convergence points $\theta^*$ as shown in Table A1. This experiment shows that the model converges to the same point in the weight space (global minimum of the loss landscape) regardless of the initialization.

Table A1: Convergence point of the model with different initialization parameters.

| Metrics | Initializations | | |
| --- | --- | --- | --- |
| | Exp 1 | Exp 2 | Exp 3 |
| $\theta_{init}$ | $[1.7640, 0.4001]$ | $[0.4967, -0.1382]$ | $[-1.5062, -0.5786]$ |
| $\theta^*$ | $[0.3554, 0.3554]$ | $[0.3554, 0.3554]$ | $[0.3554, 0.3554]$ |

We now introduce the second dataset $\mathcal{D}_2$ consisting of two data points $\{(0.5, 0.5), (0.7, 0.7)\}$ with labels $\{0, 1\}$. We train the model using $\mathcal{D}_1$ and $\mathcal{D}_2$ in four setups: (1) training only on $\mathcal{D}_1$; (2) training only on $\mathcal{D}_2$; (3) training on $\mathcal{D}_1$ followed by $\mathcal{D}_2$; (4) training only on $\mathcal{D}_2$ followed by $\mathcal{D}_1$. We present the results for this experiment in Table A2, where we observe that changing the order of the training, under the assumption of strong convexity of the loss function, changes the convergence point in the weight space, with the last training step being the determining factor.

Table A2: Convergence point of the model when trained on different datasets.

| Metrics | Datasets | | | |
| --- | --- | --- | --- | --- |
| | $\mathcal{D}_1$ | $\mathcal{D}_2$ | $\mathcal{D}_1 \rightarrow \mathcal{D}_2$ | $\mathcal{D}_2 \rightarrow \mathcal{D}_1$ |
| $\theta^*$ | $[0.3554, 0.3554]$ | $[0.2264, 0.2264]$ | $[0.2264, 0.2264]$ | $[0.3554, 0.3554]$ |

Relying on these results and the above explanation on the loss landscape and convergence points, the concept of forgetting in a strongly convex setup can be explained as follows. First, the model is trained on $\mathcal{D}_1$ and converges to its corresponding unique global minimum ($\theta_{t1}^* = \theta_1$), as the convergence of strongly convex loss using SGD is guaranteed. Next, starting from $\theta_1$, the model is trained on $\mathcal{D}_2$. Consequently, the model will converge to the unique global minimum of the second dataset's loss landscape ($\theta_{t2}^* = \theta_2$). Since the loss function is strongly convex, $\mathcal{L}(\mathcal{D}_1, \theta)$ has one global minimum which happens in $\theta_1$. As a result, the

loss value in all other points in the weight space is larger than its value at its minimizer $\theta_1$. Hence, it can be concluded that $\mathcal{L}(\mathcal{D}_1, \theta_2) > \mathcal{L}(\mathcal{D}_1, \theta_1)$ which means the second step of training increases $\mathcal{L}(\mathcal{D}_1, \theta)$. This increase in the loss of the first dataset in the sequential learning of these two datasets represents catastrophic forgetting.

## C  Implementation Details

All our experiments were conducted utilizing the PyTorch framework. We use the original implementation of Learning without Forgetting (LwF), Elastic Weight Consolidation (EWC), a modified version of EWC, Gradient Projection Memory (GPM), Scaled Gradient Projection (SGP), FOSTER, and TPL.[1]. The specifics of each algorithm's implementation are summarized in Table A3. The parameters for each algorithm such as Learning Rate (LR), Optimizer, Momentum, LR Scheduler, CL Weight, and number of Epochs are detailed.

Table A3: Table A1: Implementation Details of the Considered Algorithms for LTR benchmark.

| Algorithm | LR | Opt. | Momentum | LR Scheduler | CL Loss Weight | Epochs | Steps |
|---|---|---|---|---|---|---|---|
| LwF | 0.001 | SGD | 0.9 | - | 0.01 | 5 | 2 |
| EWC | 0.01 | SGD | 0.9 | - | 10 | 90 | 2 |
| Modified EWC | 0.01 | SGD | 0.9 | - | 1000 | 90 | 2 |
| GPM | 0.001 | SGD | 0 | Cosine Anneal LR | - | 100 | 2 |
| SGP | 0.001 | SGD | 0 | Cosine Anneal LR | - | 150 | 2 |
| FOSTER phase1 | 0.1 | SGD | 0.9 | Cosine Anneal LR | - | 170 | 4 |
| FOSTER phase2 | 0.1 | SGD | 0 | Cosine Anneal LR | - | 130 | 4 |
| TPL | 0.01 | SGD | 0.8 | Cosine Anneal LR | - | 50 | 4 |

Note that for SGP, the algorithm-specific hyperparameters are acquired through grid search as follows: $gpm_{eps} = 0.96$ and $gpm_{eps-inc} = 0.004$.

## D  Datasets

Fig. A1 illustrates the distribution of samples among different classes and the division of the dataset into the Head and Tail sections. In the case of CIFAR100-LT with $IF = 100$, the initial partition is configured such that 5% of the samples fall within the Tail and 95% in the Head section (Classes 60 to 100 are classified as Tail). For comparison purposes, the rest of the datasets follow a similar partition threshold where 60% of the classes are assigned to the Head section.

---

[1]The code for the algorithms was obtained and modified from various open-source repositories:
https://github.com/ngailapdi/LWF
https://github.com/shivamsaboo17/Overcoming-Catastrophic-forgetting-in-Neural-Networks
https://github.com/MahdiyarMM/Continual-pedestrian-detection
https://github.com/sahagobinda/GPM
https://github.com/sahagobinda/SGP
https://github.com/G-U-N/ECCV22-FOSTER
https://github.com/linhaowei1/TPL

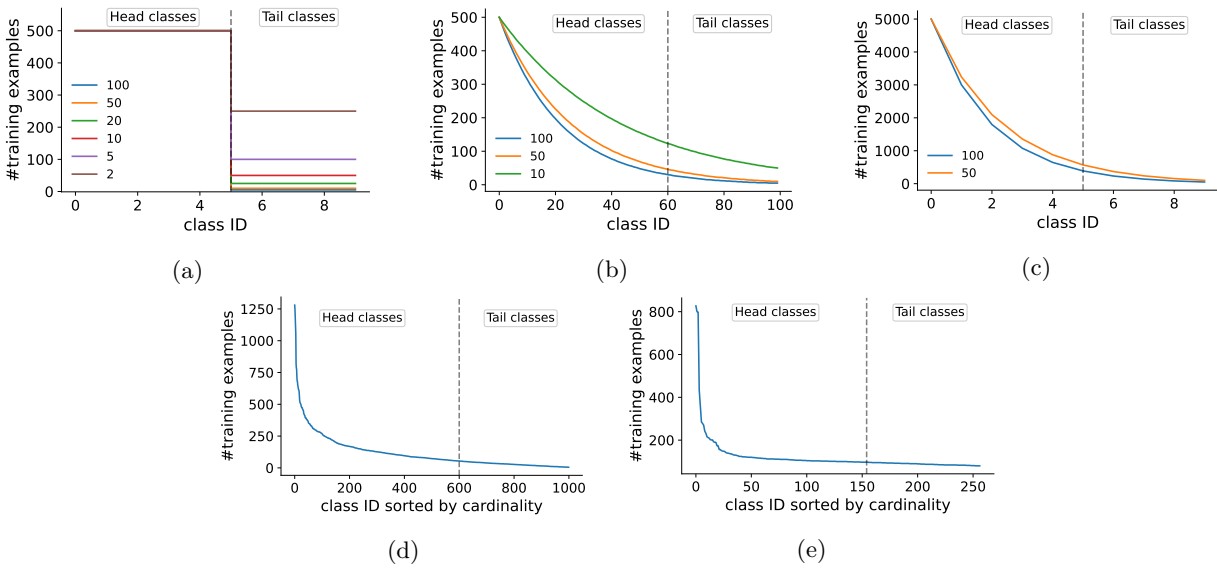

Figure A1: Class cardinality of (a) MNIST-LT, (b) CIAFR100-LT, (c) CIFAR10-LT, (d) ImageNet-LT and (e) Caltech256

An interesting phenomenon observed when training models on highly imbalanced data is the presence of artificially large weights in neurons corresponding to the Head classes (Alshammari et al., 2022). The LTR solution, WD, addresses this problem by penalizing weight growth using weight decay. One way to assess the network's ability to handle LTR is by analyzing the bias in per-class weight norms. To this end, we present the per-class weight norms of the Baseline, WD, and CLTR (SGP) models in Fig. A2.

## E    Weight Imbalance

The figure reveals a significant imbalance in the weight norms of the Baseline model, which is naively trained on the imbalanced dataset. In contrast, the WD and CLTR (SGP) models exhibit more uniform weight norms across different classes. Interestingly, although CLTR (SGP) starts with the heavily imbalanced weights of the Baseline model, it converges towards a more uniform weight distribution without any explicit penalty on weight growth. Unlike many other CL methods that restrict the plasticity of crucial weights, SGP only constrains the direction of the weight update in the weight space, enabling the model to converge to a more balanced weight distribution. This further demonstrates the effectiveness of CL in addressing LTR problem.

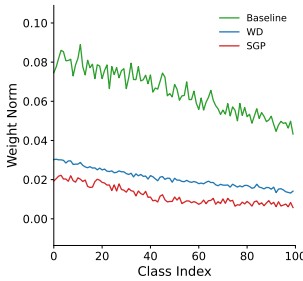

Figure A2: Per-class weight norms of the baseline, SGP, and WD.

## F    Imbalanced Binary Classification

In this work, we address the LTR problem, which inherently involves multi-class classification (Zhou et al., 2020; Zhang et al., 2022). Consequently, binary classification does not typically fall under the LTR framework and is beyond the primary scope of our study. However, we demonstrate that our proposed method, CLTR, is also applicable to binary imbalanced learning scenarios. To evaluate CLTR on this setup, we adopt a two-stage training process. First, we train the model on the Head class using one-class classification inspired by (Perera & Patel, 2019). Here the model is trained to detect the Head class samples among other unlabeled samples from an external dataset. Then, the model is trained on the Tail class samples with a replay

memory storing a few samples from the Head. We conduct these experiments on two datasets, MNIST-LT and CIFAR10-LT. For the first dataset, we select a random class from the Head and a random class from the Tail: class 4 with 1,000 samples (Head) and class 7 with 100 samples (Tail). We then train a logistic regression model on the Head, alongside 1,000 unlabeled randomly selected samples from other classes excluding the Tail as the external data. We then train the model on the Tail using CL. A replay memory with 100 samples of the Head is also employed in this stage. For CIFAR10-LT, we randomly select 5,000 samples of class 3 (Head) and 500 samples of class 5 (Tail) and train a ResNet-18 model under the same settings. We compare the performance of the model on the Tail, Head, and the balanced test set against the following models: BCE as the naive baseline trained on the entire imbalanced dataset using Binary Cross-Entropy loss; BCE (Balanced loss), where the loss associated with each class is weighted based on its size; and BCE (Balanced dataset), where the model is trained on the balanced version of the dataset in which the Head is under-sampled to ensure both classes are of the same size. We use two versions of CLTR for this experiment with EWC and SGP as the CL method. The results of this experiment are presented in Table A4, demonstrating that CLTR can effectively improve the overall performance and reduce the performance gap between the Head and Tail classes in imbalanced binary classification problems.

Table A4: The performance of CLTR on binary imbalanced classification.

| Model | MNIST-LT (IF = 10) | | | CIFAR10-LT (IF = 10) | | |
|---|---|---|---|---|---|---|
| | Acc. Minority | Acc. Majority | Acc. Overall | Acc. Minority | Acc. Majority | Acc. Overall |
| BCE | 0.0 | 99.9 | 50.0 | 10.3 | 98.3 | 54.3 |
| BCE (Balanced loss) | 91.5 | 99.2 | 95.2 | 14.5 | 95.9 | 55.2 |
| BCE (Balanced dataset) | 89.4 | 91.6 | 90.6 | 57.5 | 54.7 | 56.2 |
| CLTR (EWC) | 95.8 | 95.0 | 95.4 | 52.2 | 68.0 | 60.1 |
| CLTR (SGP) | 96.4 | 98.1 | **97.3** | 58.7 | 70.5 | **64.6** |

