# OpenReview forum: "Continual Learning for Long-Tailed Recognition"
_TMLR — Rejected by TMLR_

### Review · Reviewer_48N3 · 2024-05-27

**Summary Of Contributions:**

The authors proposed a new method to handle imbalanced datasets in terms of classification classes, which they call long tailed. Given my limited expertise on empirical research, I will mostly focus on the theoretical results. In this direction, the authors proved a series of theoretical results on imbalanced classes in the regularized logistic regression setting.

**Audience:**

Yes

**Claims And Evidence:**

Yes

**Requested Changes:**

Firstly, I would like the authors to discuss the fundamental issue of strongly convex settings. This is a setting where the optimizer will find the unique minimum in a fool proofed sense. I do not see why this theoretical result will be relevant, as the ordering of training will not affect the final outcome.

Secondly, I have a list of writing changes I would like to request the authors address. To start, I believe it is very important to define all the notations appropriately.

1a. The notations around datasets are confusing. First of all, since the authors eventually used the notations of $(x_i, y_i) \in \mathcal{D}$, the authors should define the variables $x_i, y_i$ and the spaces they live in. The author also used $\mathcal{D}^i$ there is not correspondence to the notation of $x_i, y_i$.

1b. Then the authors should define $\mathcal{D}$ as a set of these variables. Then the set $\\{ \mathcal{D}\_{c} \\}\_{c \in C}$ should define a partition of $\mathcal{D}$ based on the value of $y_i$, and $|\mathcal{D}_c|$ defines the cardinality.

1c. Then at this point, the authors should also clearly define what $\mathcal{D}\_H, \mathcal{D}\_T$ are, since the notation of $\mathcal{D}\_{c^{\text{max}}} \in \mathcal{D}\_H$ does not make sense, unless $\mathcal{D}_H$ is a set of sets of data points, which also directly conflicts with the notation $|\mathcal{D}_H|$, which would give the number of classes, not the number of data points.

1d. All of these definitions should be introduced before they are used in a proof, e.g. proof of Theorem 3.3, albeit not necessarily inside a definition environment, unless the authors want to stylistically highlight a specific definition.

1e. Other notations not introduced within the proof of Theorem 3.3 include, $\theta, P, f, \beta$. The reader should be able to easily find what all of these notations refer to.

1f. The statement of Theorem 3.3 with respect to the definition of $\delta$ is very confusing. I suggest including equation 6 in the Theorem statement.

Secondly, there are several notational issues regarding the loss function.

2a. The loss function was first introduced as a function of a set of data, e.g. $\mathcal{L}( \mathcal{D} )$. However, half way through the proof of Theorem 3.3, we saw several different versions, $\mathcal{L}(\mathcal{D}, \theta), \mathcal{L}( \beta, \theta ), \mathcal{L}(x)$. I would request the authors to please define one complete version of $\mathcal{L}, then perhaps when it is being used in a specific context, consider shortening it before a specific equation when it is not ambiguous.

2b. There are several issues with the notation $\nabla^2 \mathcal{L} (\beta, \theta) \geq \mu$. As mentioned earlier, $\beta$ is not defined, but furthermore it is not clear which derivative this is taking (i.e. with respect to $\beta$ or $\theta$). Finally, the notation of a matrix $\geq$ a scalar is not well defined. While this can be made sense of in context, I believe the authors should make the reading experience as light as possible, given there are already a lot of confusion regarding other contents.

2c. The term "regularized cross entropy loss" was first used in Assumption 3.2, but not defined at all.

Stylistically, I also have some comments, which are subjective by nature but very distracting. As a result, I also strongly recommend the authors change these.

3a. The imbalance factor is denoted $IF$, two capital letters in a math environment, without using a text or operator environment. This is problematic because it looks like a product of $I$ and $F$, as it gets naturally italicized in a math environment. I would recommend avoiding defining the ratio using two capital letters to begin with, and use a Greek letter such as $\alpha, \beta, \kappa$ (many available choices) instead. However, if the author insists on using English letters, I would recommend using either \text or define it as an operator, as commonly used in Riemannian geometry to denote grad, div, Ric etc.

3b. In Definition 3.1 and Assumption 3.2, there are non-mathematical content that appear to be a discussion. This is very distracting for the reader as we would need to read a long block of text with the attention needed to parse a careful mathematical statement, where every word and their ordering matters.

3c. Lemma 3.4 and 3.6 have a sentence pointing towards where the proof is. While this is helpful, it is not related to the mathematical content of the Lemmas. I would recommend move these to immediately after the statement of the Lemmas.

Edit: Fixed some latex render issues.

**Strengths And Weaknesses:**

Strengths: The proofs are most likely correct.

Weaknesses:

1. The result under the strongly convex setting is less relevant to the continual learning setting, since there's a unique minimum, and the order of training will not matter.

2. The mathematical writing is fairly sloppy. There are quite a few of undefined and ambiguous notation, remark styled discussion content inside a definition and assumption environment, and stylistic choices of the notation.

I will discuss these weaknesses in detail under Requested Changes. And once the authors finish addressing these basic concerns, I will continue to review the proof content, and later parts of the theoretical results.

---

> ### Author Response · Authors · 2024-08-31
>
> We would like to thank the reviewer for their insightful comments. Here we present detailed responses to each comment. In the revised manuscript, the changes related to these responses are marked by the color **blue**.
>
> > **Strong convexity.**
>
> When loss function $\mathcal{L}(.)$ is used for training a network on different datasets $\mathcal{D}\_1$ and $\mathcal{D}\_2$ with different distributions, the empirical loss landscape $\mathcal{E}$ will be functionally different, i.e., $\mathcal{E}\_{\mathcal{D}\_i}(\theta)=\left .\mathcal{L}(\theta,\mathcal{D})\right|\_{\mathcal{D}
>  = \mathcal{D}\_i}$
> , even under the strong convexity assumption. Consequently, the model will converge to a different set of parameters when trained on distinct datasets.
> Since the loss function is assumed to be strongly convex, there will be only one minima (the global minima),
> and the convergence of SGD to this point, under proper selection of the learning rate, is guaranteed (Rakhlin
> et al., 2011). As a result, regardless of the initialization, for each dataset, the model will converge to its
> corresponding global minima in the weight space. Consequently, if the model is trained on a sequence of
> different datasets, it will always converge to the global minimum corresponding to the final dataset. So the
> order of training can significantly change the outcome.
> This insight has been incorporated into the paper in Appendix B (page 21), along with a numerical example that further illustrates this concept. We have also altered our notations slightly to emphasize more on the difference between the loss landscape associated with each partition of the dataset and further highlight that training on each partition is a different optimization problem with a different loss function due to the difference in the data distributions. We now represent the loss on the data partition $\mathcal{D}\_c$ using $\mathcal{L}\_{\mathcal{D}\_c} = \mathcal{L}\_{\mathcal{D}\_c}(\theta) = \frac{1}{|\mathcal{D}\_c|}\sum\_{i=1}^{|\mathcal{D}\_c|} \ell((x\_i, y\_i), \theta)$, where $(x\_i, y\_i) \in \mathcal{D}\_c$.
>
> > **The notations around datasets**
>
> We now start our definitions in section 3.1 on Page 3 by defining $(x\_i,y\_i)$ and their corresponding space. We also replace $\mathcal{D}^i$ with $(x\_i,y\_i)$ in section 3.1 on Page 4 to enhance the clarity of the paper. Moreover, $\mathcal{D}$ and $\mathcal{D}\_c$ and their cardinalities are defined based on $(x\_i,y\_i)$ in section 3.1 on Page 4. We then explicitly define The head $\mathcal{D}\_H$ and Tail $\mathcal{D}\_T$ based on the number of classes in each subset in section 3.1 on Page 4. In section 3.1 on Page 4, we now define a complete version of the loss function over $\mathcal{D}\_c$ as $\mathcal{L}(\mathcal{D}\_c, \theta) = \frac{1}{|\mathcal{D}\_c|}\sum\_{i=1}^{|\mathcal{D}\_c|} \ell((x\_i, y\_i), \theta)$, where $(x\_i, y\_i) \in \mathcal{D}\_c$. Note that $\ell((x\_i, y\_i), \theta)$ is the loss of each individual sample. For brevity, we also use the notation of $\mathcal{L}\_{\mathcal{D}\_c} = \mathcal{L}\_{\mathcal{D}\_c}(\theta) = \mathcal{L}(\mathcal{D}\_c, \theta) $ and $\ell\_{(x\_i,y\_i)} = \ell\_{(x\_i,y\_i)}(\theta) = \ell((x\_i, y\_i), \theta)$.
> All the above definitions are provided in Section 3.1 on Page 3 before the proof of the Theorems.
>
> > **Missing notations in the proof of Theorem 3.3**
>
> $\theta$ is now defined in Assumption 3.2 on Page 4. $P(.)$ and $f(.)$ are defined after Eq. 7 on page 5. Using $\beta$ was a typo that is now fixed and $\nabla^2\mathcal{L}(\beta,\theta)$ is replaced with $\nabla^2\_{\theta}\mathcal{L}(\mathcal{D},\theta)$.
>
> > **Including equation 6 in the Theorem 3.3 statement**
>
> As per the suggestion of the reviewer, Eq. 6 is now included in the body of Theorem 3.4 on page 4.

---

> ### Author Response · Authors · 2024-08-31
>
> > **The notations around $\nabla^2$**
>
> As mentioned in the previous responses, $\nabla^2\mathcal{L}(\beta,\theta)$ is replaced with $\nabla^2\_{\theta}\mathcal{L}(\mathcal{D},\theta)$. This notation highlights that the derivative is with respect to $\theta$. As $\nabla^2\_{\theta}\mathcal{L}(\mathcal{D},\theta)$ is a matrix, we add an explanation In section 3.1 on Page 5 that this matrix is positive definite $\nabla^2\_{\theta} \mathcal{L}(\mathcal{D},\theta) \succeq \mu I$. Consequently, the Hessian of $\mathcal{L}$ is a matrix with its smallest eigenvalue being greater than $\mu$ .
>
> > **Definition of regularized cross-entropy loss**
>
> We now define "regularized cross-entropy loss'' as a cross-entropy loss with an additional $L^2$ regularization term $\frac{\mu}{2}\|\theta\|^2$ that prevents weights from growing excessively, in Assumption 3.2 on page 4.
>
> > **The notation of Imbalance Factor**
>
> To make the notation of the Imbalance Factor clearer, we now use "$\operatorname{IF}$" (Eq. 1 on page 4).
>
> > **Non-mathematical content in Definition 3.1, Assumption 3.2, and Lemmas**
>
> The non-mathematical content is now moved from the body of Definition 3.1 and Assumption 3.2 to the beginning of Section 3.1 on Page 4 and the proof of Theorem 3.4 on page 5. The references to the Lemma proofs are now happening outside the body of the lemmas, right afterward (pages 5 and 6).
>
> In the end, we would like to thank you for your thoughtful, informative, and accurate comments and feedback. We are especially very delighted with the fact that the review not only points out the issues in detail but also provides a clear path to addressing them.

---

> > ### Comment · Reviewer_48N3 · 2024-09-04
> > **Response**
> >
> > Thank you for the reply. I'm going to be honest, given the gap of time between reviewing and the discussion period, I do not remember all the details of the initial draft and my thoughts when writing the review. So forgive me if there are any inconsistencies.
> >
> > Let us start with the discussion with strong convexity. I don't remember exactly my line of thinking originally, but I do feel that strong convexity may not be the appropriate setting to study an algorithm that merely changes the order of training. I would be highly skeptical if this is representative of what happens in practice.
> >
> > In particular, my intuition is that the ordering of data breaks the symmetry of the non-convex loss landscape in the initial stage of training, and finds the broad area of the modes that it converges to. However, if the data were to be changed again, then I can imagine the training will discover newer modes, which is what leads to "forgetting." However, in the convex setting, none of these intuitions can possibly apply. So I would need more convincing that the convex setting is meaningful here.
> >
> > At the same time, given the sheer amount of mathematical notational issues and the struggle I had when reading the paper originally, I think this perhaps deserves a fresh set of eyes. A new reviewer can be a better judge on whether or not the notational clarity has improved, and hopefully will also not be so bogged down by notations.

---

> ### Author Response · Authors · 2024-09-06
>
> We would like to thank the reviewer for their diligent engagement with our work.
> In terms of the time frame of the review, kindly note that we submitted our rebuttal and revised manuscript on August 30th, adhering to the two-week window following the notification that all reviews have been received, on August 18th, in compliance with the journal's policy.
>
> Please find a detailed response to the new comments below.
>
> > **Relevance of strong convexity assumption.**
>
> To demonstrate the relevance of the assumption of strong convexity in studying sequential learning, particularly CL, please see the following references from recent works in which CL methods and catastrophic forgetting are studied under the assumption of strong convexity of the loss function, showing that this is a common assumption in the area:  Lin et al., 2024b; Wu et al., 2024; Hao et al., 2024; Cai \& Diakonikolas, 2024; Zeno et al., 2021; Bennani et al., 2020. These references are now added to Appendix B (page 21).
>
> Regarding the applicability of our theorem in practice, our empirical findings in Section 4.2 under 'Empirical support for Theorem 3.3' support our theoretical framework. Our experimental results presented in Figures 2 and 3 (page 9) demonstrate that
> the actual measured weight distances are consistently lower than the theoretical upper bounds in compliance to our theoretical findings. Additionally, our experiments in Tables 1, 2, and 3, on three commonly used LTR datasets demonstrate that CLTR is consistently among the best-performing methods.
>
> The concept of forgetting in a strongly convex setup can be explained as follows. First, the model is trained on $\mathcal{D}\_1$ and converges to its corresponding unique global minimum ($\theta^*\_{t1} = \theta\_1$), as the convergence of strongly convex loss using SGD is guaranteed. Next, starting from $\theta\_1$, the model is trained on $\mathcal{D}\_2$.
> Consequently, the model will
> converge to the unique global minimum of the second dataset's loss landscape ($\theta^*\_{t2} = \theta_2$).
> Since the loss function is strongly convex, $\mathcal{L}(\mathcal{D}\_1,\theta)$ has one global minimum which happens in $\theta\_1$. As a result, the loss value in all other points in the weight space is larger than its value at its minimizer $\theta\_1$.
> Hence, it can be concluded that $\mathcal{L}(\mathcal{D}_1,\theta_2) > \mathcal{L}(\mathcal{D}_1,\theta_1)$ which means the second step of training increases $\mathcal{L}(\mathcal{D}_1,\theta)$. This increase in the loss of the first dataset in the sequential learning of these two datasets represents catastrophic forgetting. This extended explanation is now added to Appendix B (page 21).
>
>
> > **Clarity of the mathematical notations**
>
> In response to the reviewer's suggestions, Section 3.1 now begins with the definition of the training samples and their corresponding space. To address the comment about the formal definitions of Head and Tail, we have now defined these terms based on the number of classes in each set on page 4. Following the recommendation to first present the full form of the loss function, we now formally define it before introducing its abbreviated variations for brevity in Section 3.1 (pages 3-4). Additionally, As per the request of the reviewer, to support the proof of Theorem 3.3, we have included the necessary definitions, such as the Hessian of the loss function, in Assumption 3.2 (page 4). As requested, we have also updated the notation for the imbalance factor and provided further explanation of the regularized cross-entropy loss formulation. Finally, the rest of the paper has been revised to ensure consistency with these changes.
>
> We hope these responses address your concerns and further clarify the corresponding changes in the manuscript. We would be happy to provide any further clarifications or answer any other questions that you may have.
>
> **References**
>
> Weichen Lin, Jiaxiang Chen, Ruomin Huang, and Hu Ding. An effective dynamic gradient calibration method
> for continual learning. arXiv preprint arXiv:2407.20956, 2024b.
>
> Yichen Wu, Long-Kai Huang, Renzhen Wang, Deyu Meng, and Ying Wei. Meta continual learning revisited:
> Implicitly enhancing online hessian approximation via variance reduction. ICLR, 2024.
>
> Jie Hao, Kaiyi Ji, and Mingrui Liu. Bilevel coreset selection in continual learning: A new formulation and
> algorithm. NeurIPS, 2024.
>
> Xufeng Cai and Jelena Diakonikolas. Last iterate convergence of incremental methods and applications in
> continual learning. arXiv preprint arXiv:2403.06873, 2024.
>
> Chen Zeno, Itay Golan, Elad Hoffer, and Daniel Soudry. Task-agnostic continual learning using online
> variational bayes with fixed-point updates. Neural Computation, 33(11):3139–3177, 2021.
>
> Mehdi Abbana Bennani, Thang Doan, and Masashi Sugiyama. Generalisation guarantees for continual
> learning with orthogonal gradient descent. arXiv preprint arXiv:2006.11942, 2020.

---

### Review · Reviewer_d2AJ · 2024-06-07

**Summary Of Contributions:**

In this paper, the authors propose to apply continual learning techniques for imbalanced multiclassification scenarios. The model is trained by major classes first and then finetuned by minor classes while maintaining the knowledge of major classes through continual learning.

**Audience:**

Yes

**Claims And Evidence:**

No

**Requested Changes:**

To be accepted at a top venue, this work needs significant improvements in clarity, generality, and performance.

**Strengths And Weaknesses:**

The clarity of this paper needs to be significantly improved. For instance:
1. In the introduction and related work, the authors summarized prior works of imbalanced classification, however, it is unclear what the disadvantages those methods (e.g. balancing training samples, regularizing loss or gradients) have and why the multi-stage training strategy is better.
2. The two papers -- Zhou et al. 2020 and Zhang et al. 2022 -- have been cited several times and are highly related to the proposed method, but there is no description of the methods introduced by both papers. The paper should be self-contained for such information.
3. The clarity of mathematic formulation is poor. For example, the $\lambda_f$ and $\lambda_h$ in Eq. 11 are incorrect expressions (should be $\lambda_H$ and $\lambda$ according to the context) and were copied from Eq. 12. The proof sketch of Theorem 3.8 is confusing. Eq. 14 is impossible in a continual learning setting, as a model in continual learning will never be trained on the entire dataset $\mathcal{D}$.

The novelty is incremental and the limitation is significant:
1. The authors claim that they prove a theorem to bridge continual learning and long-tail recognition (imbalanced classification), despite the correctness of their formulation, the theorem is under the assumptions that the model is logistic regression and the loss of continual learning methods is like EWC. However, only regularization-based methods in continual learning may resemble the loss of EWC, this theorem lacks generality.
2. Applying continual learning to imbalanced classification is not feasible for binary classifications, so the proposed method can only be applied to multiclass scenarios. In my experience, binary classification is a major scenario of imbalanced datasets, such as disease diagnosis, unusual events detection, etc.
3. The proposed method simply applies existing methods in continual learning to imbalanced classification, and the experimental results have not given notable improvements (Tab. 1 - 3).

---

> ### Author Response · Authors · 2024-08-30
>
> The authors would like to appreciate the review's comments and feedback. We provide a detailed response to each comment in the following posts. In the revised manuscript, the changes related to these responses are marked by the color **orange**.
>
>
> > **Disadvantage of LTR methods prior to multi-stage training**
>
> In the introduction on Page 1, we have added the following description on the disadvantages of sample-wise and loss-wise resampling  methods which have been introduced prior to multi-stage training techniques:
>
> However, both sample-wise and loss-wise balancing methods lead to increased sensitivity to variations in the tail (Wang et al., 2021c).
> It has also been shown that these methods may compromise the representational capability of the deep features learned by the model (Zhou et al. , 2020).
>
> > **Description of Zhou et al. 2020 and Zhang et al. 2022.**
>
> To further describe the method introduced in the mentioned studies, we have added the following description to Section 2 on pages 2 and 3 under Long-Tailed Recognition:
>
> Zhou et al. 2020 propose a method that addresses
> the training of the encoder and classifier separately through a novel cumulative learning strategy that initially
> focuses on universal patterns before progressively concentrating on the Tail.
>  Zhang et al. 2022 introduce a new method called Self-supervised Aggregation of Diverse Experts, which trains multiple experts on a long-tailed dataset to manage various class distributions and uses self-supervision at test time to combine these experts for unknown class distributions.
>
> > **$\lambda_f$ and $\lambda_h$ in Eq. 11**
>
> We have fixed Eq. 11 on page 5 by replacing $\lambda_f$ and $\lambda_h$ with $\lambda_H$ and $\lambda$, respectively.
>
>
> > **The proof sketch of Theorem 3.8**
>
> The proof sketch of Theorem 3.8 on page 6 is revised to ensure its clarity. We now describe the main steps that we took to prove this theorem in more detail.
>
> > **Eq. 14 and CL**
>
> Note that in Theorem 3.10 we use the loss of the entire dataset as a theoretical measure for the upper bound, and the model is indeed not trained on the whole dataset. We have now added a further explanation on this matter at the end of the proof sketch of Theorem 3.10 on page 7 to ensure this is clear in the paper.
>
>
> > **Lacking generality**
>
> Kindly note that EWC was only used as an **example** of a CL method for theoretical demonstration of the effectiveness of the proposed framework. Our reformulation of LTR as a general form of CL was independent of EWC. Our formulation (in Theorems 3.4 and 3.8) ensures that **any solution** crafted for CL, even beyond regularization-based methods, can be applied to the LTR problem. We have also demonstrated the applicability of an extensive range of CL solutions (both regularization-based and non-regularization-based) on different architectures for addressing LTR problems in Tables 1 to 3 (page 10), showcasing the generalizability of our proposed framework beyond a certain kind of CL solution.
>
>
> > **Applicability to binary classification**
>
> In this work, we address the **LTR problem** which is a multi-class problem (Zhou et al. 2020 and Zhang et al. 2022). Accordingly, binary classification does not fall under this category and is out of the scope of this work, following prior works such as:
> (Cui et al., 2019a, Lin et al., 2017a, Kang et al., 2019, Zhou et al., 2020, Tang et al., 2020a, He et al., 2021, Samuel and Chechik, 2021, Alshammari et al., 2022).
> Finally, all the well-known LTR benchmarks contain more than two classes (CIFAR10-LT, CIFAR100-LT, ImageNet-LT, etc).
>
>
> > **Novelty and notable improvements.**
>
> Regarding novelty, kindly note that we theoretically prove that any off-the-shelf CL method can be applied for LTR, for the first time. Next, we empirically demonstrate that applying CL methods on LTR can achieve superior or competitive results to methods that have been specifically designed for LTR. Our theoretical framework can have considerable implications for the field of LTR, while our empirical results set new state-of-the-art performances across three datasets (Tables 1 through 3 on page 10). Finally, we also show that our framework can have significant implications when applied to long-tail class-incremental learning (LT-CIL) problems and outperform the state-of-the-art methods by notable margins (Table 4 on page 10).

---

> > ### Comment · Reviewer_d2AJ · 2024-10-01
> > **Thanks for authors' response**
> >
> > Thanks to the authors for amending the manuscript according to my comments. I'm sorry that I don't have enough time to read the revised paper through. The mathematical results still look a bit messy to me. The conclusion is based on strong assumptions about convexity and linearity of the model. The authors claim that it is general to any solution in CL. It is unclear how this can be achieved, as there are quite diverse methods in CL. Regarding the limitations of applications, binary classification is common in LTR scenarios, and the work is proposed for LTR, so in my understanding, it is an actual imitation in practice.

---

> > > ### Author Response · Authors · 2024-10-02
> > >
> > > We would like to thank the reviewer for their continued engagement. We understand that you are busy and unable to review the paper any further, and appreciate the time you have already put in. We just wanted to take this opportunity and provide a few points for further clarification:
> > >
> > > - The math: We carefully revised the initial version as per your comments. We would appreciate it if you could kindly point out anything that we may have missed.
> > >
> > > - Strong convexity: This is a standard assumption in CL. Recent references are provided in Appendix B (page 21).
> > >
> > > - Linearity: We used a logistic regression model as the classifier, which is non-linear, as noted in Assumption 3.2 (page 4).
> > >
> > > - Applicability to CL solutions: As one of our contributions in this paper, we have reformulated the LTR problem as a CL problem. Hence, any method capable of addressing CL can indeed address LTR as per our proposed framework. This concept is explained in detail in Section 3.2 on pages 6-7.
> > >
> > > - Binary classification and LTR:
> > > As discussed in our earlier response, as per the definition of LTR by (Zhou et al. 2020 and Zhang et al. 2022), LTR is an inherently multi-class problem (binary problems can not have a long tail by definition). Accordingly, all LTR benchmarks and datasets are multi-class (See Table 1 in (Fu, Yu, et al. 2022)).
> > >
> > > References
> > >
> > > Fu, Yu, et al. "Long-tailed visual recognition with deep models: A methodological survey and evaluation." Neurocomputing 509 (2022): 290-309.

---

> > > ### Author Response · Authors · 2024-10-04
> > >
> > > We have now further investigated the applicability of the proposed method to binary imbalanced learning. To this end, we conducted additional experiments on CIFAR100-LT and MNIST-LT by implementing a two-stage training process. We first train the model on the Head class using one-class classification inspired by (Pramuditha & Patel, 2019). Here the model is trained to detect the Head class samples among other unlabeled samples from an external dataset. Then, the model is trained on the Tail class samples with a replay memory storing a few samples from the Head. The results are presented in the table below:
> > >
> > >
> > > |       **Model**                  |  |       **MNIST-LT**        |               |  |          **CIFAR10-LT**     |               |
> > > |----------------------|----------------------------|---------------|---------------|------------------------------|---------------|---------------|
> > > |          | **Acc. Minority**         | **Acc. Majority** | **Acc. Overall** | **Acc. Minority**         | **Acc. Majority** | **Acc. Overall** |
> > > | **BCE**              | 0.0                        | 99.9          | 50.0          | 10.3                        | 98.3          | 54.3          |
> > > | **BCE (Balanced loss)** | 91.5                    | 99.2          | 95.2          | 14.5                        | 95.9          | 55.2          |
> > > | **BCE (Balanced dataset)** | 89.4                | 91.6          | 90.6          | 57.5                        | 54.7          | 56.2          |
> > > | **CLTR (EWC)**        | 95.8                      | 95.0          | 95.4          | 52.2                        | 68.0          | 60.1          |
> > > | **CLTR (SGP)**        | 96.4                      | 98.1          | **97.3**          | 58.7                        | 70.5          |**64.6**          |
> > >
> > > These experiments demonstrate that CLTR can effectively improve the performance of models trained on imbalanced datasets consisting of only two classes. These experiments and the results have now been added to Appendix F (pages 24 and 25) of the paper.
> > >
> > > **References**
> > >
> > > Perera, Pramuditha, and Vishal M. Patel. "Learning deep features for one-class classification." IEEE Transactions on Image Processing 28.11 (2019): 5450-5463.

---

### Review · Reviewer_MbSy · 2024-08-19

**Summary Of Contributions:**

The authors propose a framework named CLTR to tackle the problem of learning class-imbalanced data. They address the long-tailed recognition problem using continual learning method, and provide theoretical and experimental evidence. Sufficient experiments show that CLTR gives competitive results against state-of-arts mothods.

**Audience:**

Yes

**Claims And Evidence:**

Yes

**Requested Changes:**

1. in Theorem 3.3, what are the "strong convexity parameters" ($\mu$ and $\mu_H$) and how they are obtained?
2. From Eq. (4) to Eq. (5) where the authors replace $\frac{|D_H|}{|D|}$ with $\gamma$, the author can give more evidence or explaination on this step.
3. In Eq. (7), how is the coefficient of the regularization is $\frac{\mu}{2}$ obtained?
4. In Theorem 3.9, "for all $s < S-i$, where $s$ is the number of past training ...", isn't $i$ indicating the number of past training?
5. The authors are suggested to move the algorithm of their framework (Appendix B) into the main pages. The procedure of generating multiple partitions are very important, in my opinion.
6. In the framework algorithm, the authors should explain the meaning of symbols such as $D_i$, $L$ and $N$. I understand $L$ is the set of labels and $N$ is the number of partitions, but it took me a while to guess it. Also, shouldn't $N$ a part of the input?
7. In Eq. (13) where the authors likening the LTR problem to CL, I am not very convinced about $\mathcal{Y_t} = D_H$ and $\mathcal{Y}_{t+1} = D_T$ after multiple steps.
8. In Sec. 4 experiment setting, the authors should clearify the number of partitions they use to get the results in Tabel 1,2,3 in the main pages.

**Strengths And Weaknesses:**

**Strengths**:  The authors provide mathematical proof on why they can address the problem of insufficient learning of long-tailed classes by continual learning techniques. Also, they conduct experiments to support their mathematical proof and the effectiveness of their framework.

**Weakness**: The current writting of Section 3 is a little hard to follow. Some places could be given more explainations to make their proposal more convincing. The procedure of the CLTR framework (Appendix B) should be moved to the main pages as it is very important.

---

> ### Author Response · Authors · 2024-08-30
>
> We thank the reviewer for their detailed and thoughtful feedback. Here, we provide a response to all of the comments and questions. In the revised manuscript, the changes related to these responses are marked by the color **Red**.
>
> > **More explanation in Section 3.**
>
> To enhance the clarity and readability of Section 3 in our paper, we have revised some of the notations and provided additional explanations of the mathematical expressions (highlighted in Blue and Red). We have extended our explanations in Sections 3.1 and 3.2 (pages 3 to 8), as well.
>
> > **Move Algorithm 1 to the main text.**
>
> We have now moved Algorithm 1 to Section 3.2 (page 7), as per your suggestion.
>
>
> > **Strong Convexity Parameters $\mu$ and $\mu_h$.**
>
> $\mu$ and $\mu_h$ represent the extent of strong convexity of $\mathcal{L}$ and $\mathcal{L}_H$. $\mu$ is defined as the lower bound of the eigenvalues of the Hessian matrix (as explained after Eq. 7 on page 5). This value is determined based on the loss formulation. As defined in Eq. 7, we use the cross-entropy loss with an additional $L^2$ regularization term: $\mathcal{L}=CE + \frac{\mu}{2} \|\theta\|^2$. As the Hessian of CE is positive semi-definitive, the lower bound of the Hessian of the loss will be the coefficient of the regularization $\mu$, hence, $\nabla^2 f(x) \succeq \mu I$. This reflects the extent of strong convexity of the loss function. This description has been added to the manuscript in the paragraph following Eq. 7 (on page 5). Note that in the case that the same coefficient for L2 regularization is employed for both phases of the training, the upper bound in Eq. 10 can be further simplified to $  \frac {2\delta}{\mu}$. We have explained this simplification in the paragraph after Eq. 10 (page 6).
>
> > **More explanation on $\gamma$.**
>
> We define $\gamma$ as $ \frac{|\mathcal{D}_H|}{|\mathcal{D}|}$. Consequently,  $1-\gamma= 1-\frac{|\mathcal{D}_H|}{|\mathcal{D}|}=\frac{|\mathcal{D}|-|\mathcal{D}_H|}{|\mathcal{D}|}$. As $|\mathcal{D}|=|\mathcal{D}_H|+|\mathcal{D}_T|$, it yields that $1-\gamma= \frac{|\mathcal{D}_T|}{|\mathcal{D}|}$. By plugging $\gamma$ into Eq. 4 we can derive Eq. 5. Furthermore, as $\operatorname{IF}$ is defined a $\frac{|\mathcal{D}_H|}{|\mathcal{D}_T|}$, we conclude that $\gamma=\frac{\operatorname{IF}}{1+\operatorname{IF}}$ which falls within the range of $[0.5,1)$. This additional explanation has been added to the manuscript before and after Eq. 5 (pages 4 and 5).
>
>
>
> > **$\frac{\mu}{2}$ in Eq. 17.**
>
> The value of $\mu$ is a training hyper-parameter determined by the user, usually through hyper-parameter tuning, grid search, or similar approaches. We have now expanded the explanation about $\mu$ in the paragraph after Eq. 7 (page 5). We have also demonstrated the impact of different values of $\mu$ on the network in Fig. 2 and Fig. 3 (page 9). Our results in Fig. 3 illustrate that our theorem holds for different values of $\mu$.  In this experiment, we changed the value of $\mu$ in different $\operatorname{IF}$s and compared it against the predicted upper bound, demonstrating the actual weight difference values are below our predicted upper bounds in all cases. Further explanations of these experiments are also presented in Section 4 (page 9).
>
> > **The number of past training epochs in Theorem 3.9.**
>
> The model is trained for $i$ epochs on the Head. So $i$ denotes the number of epochs in the first phase of the training. From this point, we train the model using CLTR for another $S$ epochs. Theorem 3.9 proves that using CLTR yields a lower value of loss than fine-tuning in all the steps of training after the initial $i$ steps, denoted by $s$. This theorem holds for any epoch number between the initial steps $i$ and the total number of steps in both phases $S$. Hence, $s$ denotes the number of past epochs in the second phase of the training, while $i$ is the number of past epochs in the first phase. We have clarified the difference between $i$, $s$, and $S$ immediately before and after Eq. 14 (page 8).
>
>
> > **Provide the notations in Algorithm 1.**
>
> We have now revised Algorithm 1 (page 7) and introduced all the notations including the number of partitions $N$, the cardinality of each class $|\mathcal{D}_i|$, and the group of partition boundaries $L$. We have also included $N$ in the inputs.

---

> ### Author Response · Authors · 2024-08-30
>
> > **Regarding $\mathcal{Y}_t=\mathcal{D}_H$ and $\mathcal{Y}\_{t+1}=\mathcal{D}_T$ in Eq. 13.**
>
> Relying on Theorem 3.8, which proves that Theorem 3.3 holds for any number of steps, we have now slightly revised Eq. 13 (page 7) and its previous two paragraphs to show that the reformulation of LTR as a CL problem holds for multiple steps.
>
> Following (Prabhu et al., 2020), a general CL problem can be formulated as a model exposed to a stream of $N$ incoming training datasets $\mathcal{D}\_{\mathcal{Y}_t}= \\{ (x_i,y_i)|y_i\in\mathcal{Y}\_t \\}$ for $1\leq t\leq N$,
> where $\mathcal{Y}\_t$ is the corresponding set of labels. Up to the current timestep $t$, the set of labels $\bigcup\_{i=1}^t \mathcal{Y}\_i$ in dataset $\bigcup\_{i=1}^t \mathcal{D}\_{\mathcal{Y}\_i}$ has been previously used in training of the network.
> The objective at the next timestep $t+1$ is to find a mapping $f\_{\theta}:x\rightarrow y$ that accurately maps sample $x$ to $\bigcup\_{i=1}^t \mathcal{Y}\_i \cup \mathcal{Y}\_{t+1}$, where $\mathcal{Y}\_{t+1}$ is the set of new unseen labels in the incoming new dataset $\mathcal{D}\_{\mathcal{Y}\_{t+1}}=\\{ (x_i,y_i)|y\_i\in \mathcal{Y}\_{t+1} \\}$. Therefore the ultimate objective of CL is to find an accurate mapping $f\_{\theta}:x\rightarrow y$ for all $(x,y)\in \bigcup\_{i=1}^N \mathcal{D}\_{\mathcal{Y}_i}$.
>
> Consider dataset $\mathcal{D}$ under the LTR setup (Definition 3.1) divided into $N$ partitions with substantially different sizes sorted based on cardinality such that $\mathcal{D} = \bigcup\_{i=1}^{N} \mathcal{D}^{i}$ and $|\mathcal{D}^{i}| >> |\mathcal{D}^{i+1}|$. We have shown in Theorem 3.3 and Theorem 3.8 that $\theta^*$ (the weights of the model when trained on the entire dataset) will be very close to $\theta^*\_1$, which is the weights of the model when it is only trained on $\mathcal{D}\_1$ (the largest partition of the dataset). As a result, the model after training on $\mathcal{D}$ can be considered as $f\_{\theta^*\_{1}}:x\rightarrow y$ for all $(x,y)\in \mathcal{D}\_1$.
> On the other hand, following Definition 3.1, the objective of LTR is to learn $f_\{\theta}:x\rightarrow y$ for all $(x,y)\in \mathcal{D} = \bigcup\_{i=1}^{N} \mathcal{D}^i$.
> Hence, additional training steps are required for the model to further learn the rest of the partitions of the dataset ($\bigcup\_{i=2}^{N} \mathcal{D}^i$).
> Thus, if we consider each of the partitions of the LTR dataset ($\mathcal{D}^i$ for $1\leq i \leq N$) as an incoming CL dataset ($\mathcal{D}\_{\mathcal{Y}\_t}$ for $1\leq t \leq N$), the objective of the LTR problem would be equivalent to the objective of CL, which is to estimate $f\_{\theta}$:
> \begin{equation}
> f\_{\theta}:x\rightarrow y \quad\ s.t. \quad\ (x,y)\in \bigcup\_{t=1}^{N} \mathcal{D}\_{\mathcal{Y}\_t} \quad \text{and} \quad \mathcal{D}\_{\mathcal{Y}\_1}=\mathcal{D}^1,\ \mathcal{D}\_{\mathcal{Y}\_2}=\mathcal{D}^2,\ \dots ,\ \mathcal{D}\_{\mathcal{Y}\_N}=\mathcal{D}^N.
> \end{equation}
> Thus, our proposed approach unifies the two domains so that an LTR problem can be treated as a CL problem.
>
>
> This discussion is now included in the paragraph before Eq. 13 (page 7).
>
>
> > **Clarifying the number of partitions in the main text.**
>
> As per your suggestion, we have now moved the information on the number of partitions for each CL method from the appendix to the main text in Section 4.1 (page 8) under Implementation Details.

---

> > ### Comment · Reviewer_MbSy · 2024-09-04
> > **Responce To Authors**
> >
> > Thanks for the authors' reply. I carefully reviewed the changes made by the authors and believe most of my questions received clear explaination.
> >
> > However, the writing of Section 3 still has problematic and unclear places:
> > 1. in the begining of page 4,
> > > " ... $D_H = \\{(x_i, x_j) \in D: y_i <= c_K\\}$ ... where $c_k$ denotes how many classes belong to each set"
> >
> > First, the case for $c_k$ is not unified. Second, the later one should be $D_T$ not $D_H$. Third, the formal definitions of $D_H$ and $D_T$ are inconsistent with its context and should be revised again. For example, if there are three subset $D_{y_1}, D_{y_2}, D_{y_3}$ with labels $y_1, y_2, y_3$, and their sizes meet $|D_{y_1}| >> |D_{y_3}| >> |D_{y_2}|$. Then, from my understanding of the paper, $D_H$ can be $\\{D_{y_1} \cup D_{y_3}\\}$ and $D_T$ be $\\{D_{y_2}\\}$. However, by current definitions, if $c_k = 2$, then $D_H = \\{D_{y_1} \cup D_{y_2}\\}$ and $D_T= \\{D_{y_2}\\}$. I may take it wrong somewhere. But the overall description and definitions here are too complicated and hard to follow.
> >
> > 2. The readability of Proof for Theorem 3.3 is not good. The introduction of Lemma 3.4 disruptes my thoughts before the proof is finished.

---

> > > ### Author Response · Authors · 2024-09-06
> > >
> > > We would like to thank the reviewer for engaging with our submission. We are particularly delighted that our rebuttal and revision have addressed most of your questions. Please find the detailed response to your recent comments below:
> > >
> > >
> > > > **Notational issues with $c_k$ and $\mathcal{D}_T$.**
> > >
> > > To enhance the clarity of section 3, we have now revised the paper (top of page 4) and replaced $c_K$ with  $c_k$, which is more consistent with the defined label space $\\{1, \ldots, k\\}$. We have also fixed the typo regarding $\mathcal{D}_T$.
> > >
> > >
> > > > **Consistency in the formal definition of $\mathcal{D}_H$ and $\mathcal{D}_T$.**
> > >
> > > We are following (Hong et al., 2024) for the formulation of Head and Tail. This reference is now added to the manuscript in section 3.1 on page 3 for further clarification. This definition ensures that all the classes in the Head are Larger than all the classes in the Tail. Hence, the only acceptable partitioning of data in the provided example is $\mathcal{D}\_{H}=\mathcal{D}\_{y1}\cup\mathcal{D}\_{y2}$ and  $\mathcal{D}\_{T}=\mathcal{D}\_{y3}$. In our work, we are avoiding grouping $\mathcal{D}\_{y1}$ and $\mathcal{D}\_{y3}$ together as the Head since this partitioning will increase the imbalance factor within the Head itself compared to the case that $\mathcal{D}\_{y1}$ and $\mathcal{D}\_{y2}$ are considered as Head. This increase in the imbalance factor leads to sub-optimal performance of the model in learning the smaller class within the Head. To enhance the consistency of the definition of the Head and Tail, we now further emphasize the sorting of the classes based on their cardinality before assigning them to Head or Tail in the following parts of the paper:  (1) line 3 of Algorithm 1, (2) line 1 of page 4, and (3) on page 6, before Theorem 3.8.
> > >
> > >
> > > > **Readability of Proof for Theorem 3.3.**
> > >
> > > To enhance the readability of the proof, we now introduce Lemma 3.4 before starting the proof on page 4. Then we refer to this lemma on page 5 within the proof to avoid breaking the flow.
> > >
> > > We hope our responses have addressed your concerns and questions. We would be happy to provide any further clarifications or answer any other questions that you may have.

---

### Author Response · Authors · 2024-08-30

We are sincerely grateful to the reviewers for dedicating their time and offering valuable feedback. It is encouraging to receive positive and engaging remarks. We are glad to know that reviewers find the paper valuable as the proposed solution is new (Reviewer 48N3), provides both theoretical and sufficient experimental evidence to support the framework’s effectiveness (Reviewer MbSy), and achieves competitive results against state-of-the-art LTR methods (Reviewer MbSy). Moreover, the mathematical proofs have been appreciated (Reviewers MbSy, 48N3, and d2AJ). We have carefully addressed all the concerns raised by the reviewers under the individual response section. Following, we provide a summary of our responses:

— **Clarifications and Revisions**: We have provided more explanation to further clarify Section 3 of our paper, ensuring more clarity and better flow.

— **Notational Revisions**: We have revised the notations where necessary, to improve the readability and accuracy of the mathematical expressions.

— **Enhanced Discussion on Related Works**: We have expanded our discussions in the related works section to elaborate more on some of the prior works and to include a broader spectrum of relevant studies, enhancing the contextual foundation of our work.

We believe these changes substantially strengthen our paper and address the valuable feedback provided by the reviewers. We would be happy to answer any follow-up questions should they arise.

---

### Decision · Action_Editor_dzzf · 2024-11-25

**Recommendation:** Reject

**Comment:**

Even the most positive reviewer mentioned that they were 55/45% split between acceptance and rejection, with the other two reviewers either leaning to or strongly recommending rejection.

**Audience:**

Yes, the paper would be relevant to TMLR's audience.

**Claims And Evidence:**

The paper presents a framework for addressing long-tailed recognition via continual learning. The paper includes substantial theoretical contributions, such as proofs linking continual learning and long-tailed recognition, and experimental results that show strong performance. However, reviewers expressed significant concerns with clarity, substantial issues with the mathematical notation, and issues with some of the assumptions underlying the theoretical results (e.g., strong convexity, which may not fit practical settings). The authors provided a substantial revision to address the notation issues, but multiple reviewers mentioned that they still had concerns with the notation and clarity throughout. (Note that the author's private concerns mentioned were considered in preparing this meta-review, but multiple reviewers still felt that there were issues.) Even the most positive reviewer mentioned that they were only borderline in recommending acceptance, and mentioned ongoing clarity issues. Based on this, it seems that the paper would benefit from another round of significant revisions before it is ready for publication.

**Resubmission Of Major Revision:**

The authors may consider submitting a major revision at a later time.